# A Sign-aware Graph Transformer with Prototypical Objectives for Signed Link Prediction in Bipartite Graphs

## Abstract

Signed link prediction focused on bipartite graphs is a fundamental task with wide-ranging applications, yet it poses significant challenges. Current Graph Neural Networks are inherently local due to their message-passing nature, preventing them from capturing the long-range dependencies crucial for accurate prediction. Furthermore, they often fail to model complex real-world data distributions characterized by severe class imbalance and rich intra-class multimodality. To overcome these limitations, we propose the Hierarchical Prototypical Contrastive Sign-aware Graph Transformer (HPC-SGT), designed specifically for the bipartite setting. At its core, our framework features a Sign-aware Graph Transformer that operates on the line graph dual, leveraging novel spectral and motif-based inductive priors to learn structurally-aware global representations. This expressive encoder is optimized via a hierarchical prototypical objective, which learns a geometrically structured embedding space. It couples a class-balanced contrastive loss to robustly handle data imbalance with clustering and separation regularizers to explicitly model multi-modal class structures. The framework is unified by a cross-view consistency mechanism that grounds the learned semantic representations in the graph's foundational topology, bridging the structure-semantics gap. Extensive experiments on challenging benchmarks, including scenarios with severe class imbalance, show that HPC-SGT significantly outperforms a wide range of state-of-the-art methods. Ablation studies further validate the contribution of each component, establishing HPC-SGT as a new, powerful, and principled solution for signed link prediction. Our code is available in the supplementary materials.

## 1 Introduction

Signed link prediction focused on bipartite graphs constitutes a fundamental research problem in network science and machine learning Koren et al. (2009); Zhao et al. (2015); Song et al. (2015), with profound implications for a multitude of real-world systems, as shown in Figure 1. These graphs, which model interactions between two distinct sets of entities—such as users and items in e-commerce Lin et al. (2024); Tang et al. (2016); Arrar et al. (2024), voters and bills in legislative systems Maier & Simovici (2022); Yin et al. (2019); Guo et al. (2025), or individuals and groups in

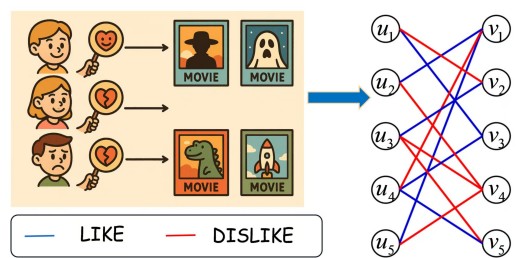

Figure 1: An illustrative example of the user-movie rating interaction in bipartite graphs.

social networks—are often endowed with signs (positive or negative) that encode the nature of the relationship, e.g., like versus dislike, or trust versus distrust. The ability to accurately forecast the sign of a new or unobserved link is paramount for applications ranging from personalized recom-

mendation Wang et al. (2025) and fraud detection to maintaining the integrity of online communities Braunhofer et al. (2015); Massa & Avesani (2007); Chen et al. (2024a).

While initial approaches relied on social theories Leskovec et al. (2010) or handcrafted features Fu et al. (2021), the field has shifted towards Graph Neural Networks (GNNs) Wu et al. (2020). Pioneering methods like SGCN Derr et al. (2018), attention-based SiGAT Huang et al. (2019), and contrastive SGCL Shu et al. (2021) have adapted message-passing to signed networks. However, these methods face two fundamental limitations. First, their reliance on iterative message-passing restricts their receptive field, rendering them ill-equipped to model long-range dependencies Zhang et al. (2020); Wang & Wu (2024); Hang et al. (2024). Second, standard objectives often fail to address complex real-world distributions, such as severe class imbalance and intra-class multimodality.

To address these limitations, we propose the Hierarchical Prototypical Contrastive Sign-aware Graph Transformer (HPC-SGT). Our Sign-aware Graph Transformer operates on the line graph dual, enabling global topological reasoning to overcome GNN locality. It incorporates graph-native inductive priors, specifically spectral balance and local motifs, directly into the attention mechanism to capture global, structurally-principled representations. We optimize this with a hierarchical prototypical objective designed to handle class imbalance and intra-class multimodality through geometric regularizers . Finally, a cross-view consistency mechanism bridges the structure-semantics gap, ensuring topological fidelity.

Extensive experiments on four benchmarks demonstrate that HPC-SGT significantly outperforms state-of-the-art baselines, particularly GNN and Transformer competitors, validating our global structurally-aware architecture. Furthermore, ablation studies confirm the essential role of our graph-native inductive priors, as their removal leads to substantial performance drops. The contributions of this work are threefold:

- To resolve the inherent locality default of existing GNNs, we propose a Sign-aware Graph Transformer operating on the line graph. By integrating novel spectral and motif-based inductive priors, it directly captures long-range signed dependencies and global structural balance that are typically inaccessible to local message-passing frameworks.

- To tackle the dual challenges of severe class imbalance and intra-class multimodality, we design a hierarchical prototypical objective. Unlike standard discriminative losses, this probabilistic framework maps links to diverse semantic prototypes, ensuring that minority classes are not submerged and that complex, non-Gaussian interaction modes are effectively modeled.

- To mitigate the structure-semantics gap in deep encoders, we introduce a cross-view consistency mechanism. This regularizer bridges the learned semantic representations with the foundational graph topology, ensuring topological fidelity and preventing the model from overfitting to spurious patterns.

## 2 RELATED WORK

### 2.1 SIGNED BIPARTITE GRAPHS AND LINK PREDICTION

Signed graphs have gained considerable attention due to their significance in social networks and recommender systems Guo et al. (2020); Chen et al. (2020). The presence of both positive and negative links enriches these graphs with complex relational dynamics, making them a valuable resource for tasks such as signed link prediction, node classification, and community detection. Signed Graph Representation Learning (SGRL) has been proposed as an effective approach to capture the intricate patterns in signed graphs and better understand the coexistence of positive and negative relationships Wang et al. (2020); Shu et al. (2021). Early SGRL methods focused on random walk strategies and matrix factorization. Random walk-based approaches like DeepWalk Perozzi et al. (2014) and node2vec Grover & Leskovec (2016) capture node proximity probabilistically, while matrix factorization Koren (2009) models signed interactions by decomposing adjacency matrices. As deep learning advanced, SiNE Wang et al. (2017) combined triangle motifs and balance theory to address positive/negative relationships. SGCN Derr et al. (2018) extended GCNs with balance theory for multi-hop signed link prediction. Further developments, such as SiGAT Huang et al. (2019) and SNEA Li et al. (2020), incorporated graph attention mechanisms, allowing more flexible weighting

of node interactions. Recent methods, including SDGNN Huang et al. (2021b), SBGCL Zhang et al. (2023), and Trans-CGL Lin et al. (2023), leverage contrastive learning to enhance the robustness of signed graph representations Qin et al. (2025a).

Despite these advances, modeling balance theory in bipartite graphs remains a challenge due to its high space and time complexities, which become impractical as the graph size grows. Consequently, while these methods improve the prediction of link signs, they still struggle with scalability and efficiency in handling vast signed graphs Ortega et al. (2018); Lin et al. (2025); Qin et al. (2025b).

## 2.2 Transformers and Line Graphs

To overcome the locality issue of GNNs, recent research has turned to the Graph Transformer architecture Li et al. (2024); Chen et al. (2024b); Zhao et al. (2025). Its global self-attention mechanism theoretically allows every node to interact with every other node, making it a promising candidate for capturing long-range dependencies. However, standard Transformers are topology-agnostic, and their effectiveness on graphs is highly dependent on the injection of explicit structural and positional encodings to make the attention mechanism aware of the underlying graph structure—a challenge our methodology directly addresses.

Parallel to this, the line graph transformation has emerged as a powerful technique for link-level tasks Xing & Makrehchi (2024). By converting edges from the original graph into nodes in a new graph, the line graph reframes signed link prediction as a node classification problem. This enables node-centric architectures like GNNs or Transformers to directly model the interactions between links. While these advanced concepts are powerful individually, a unified framework that synergistically combines a structure-aware Graph Transformer on the line graph with learning objectives tailored for the complex distributions of signed links remains an open challenge.

## 3 Preliminary

A signed bipartite graph is denoted as $G = (\mathcal{V}, \mathcal{E}, s)$, where the vertex set $\mathcal{V} = U \cup V$ consists of two disjoint partitions of nodes, such as users $U$ and items $V$. The edge set $\mathcal{E} \subseteq U \times V$ represents the interactions between these two sets of nodes. The sign function $s : \mathcal{E} \rightarrow \{+1, -1\}$ assigns a positive (e.g., like, purchase) or negative (e.g., dislike, negative review) sign to each interaction, where the set of all edges can be partitioned into positive and negative sets, $\mathcal{E} = \mathcal{E}^+ \cup \mathcal{E}^-$ with $\mathcal{E}^+ \cap \mathcal{E}^- = \emptyset$ Guo et al. (2020); Chen et al. (2020). The task of signed link prediction in this context assumes that the full edge set $\mathcal{E}$ is partitioned into a set of observed edges, $\mathcal{E}_{\text{obs}}$, for which the signs are known, and a set of target edges, $\mathcal{E}_{\text{unk}}$, for which the signs are withheld for evaluation. Given the graph structure $(\mathcal{V}, \mathcal{E})$ and the known signs on $\mathcal{E}_{\text{obs}}$, the objective is to learn a predictive function $f$ that infers the sign $y_{uv} \in \{+1, -1\}$ for each target edge $(u, v) \in \mathcal{E}_{\text{unk}}$. This is typically achieved by learning low-dimensional embeddings for all nodes that encode the complex structural patterns and sign information within the graph Perozzi et al. (2014).

## 4 Methodology

In this section, we present the technical details of our proposed framework, the Hierarchical Prototypical Contrastive Sign-aware Graph Transformer (HPC-SGT). Our approach is built upon a synergistic system of three core innovations designed to overcome the key limitations of existing methods in signed link prediction. We begin by detailing the architecture of our Sign-aware Graph Transformer, which operates on the line graph and incorporates novel inductive priors to capture global, structurally-aware representations. Next, we describe our Hierarchical Prototypical Learning Objective, a unified framework designed to handle both class imbalance and intra-class multimodality. Finally, we introduce our Cross-View Consistency mechanism, a principled regularizer that ensures the learned representations are topologically faithful.

### 4.1 Sign-aware Graph Transformer for Global Link Representation

To transcend the inherent locality of conventional GNNs, we propose a Sign-aware Graph Transformer (SGT) that operates on the line graph dual. This core component is distinguished by its

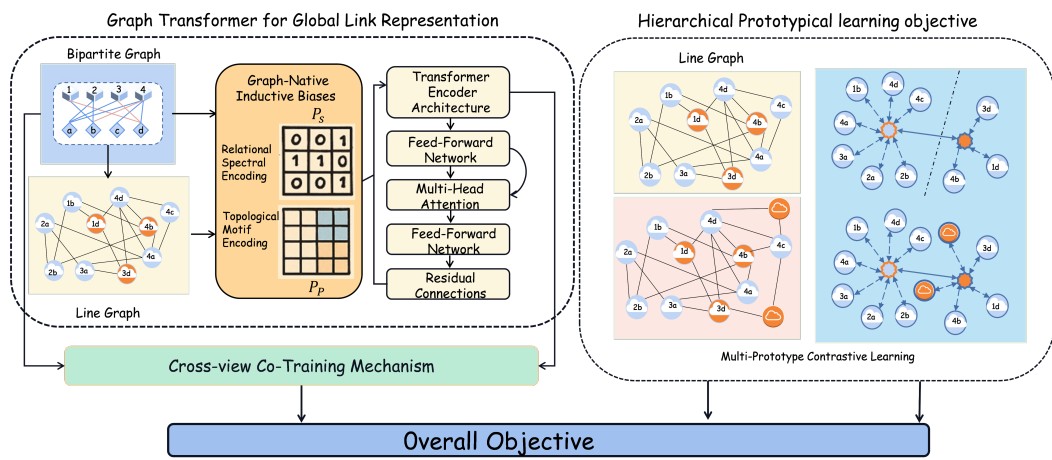

Figure 2: The integrated architecture of HPC-SGT. The framework learns global link representations using a Sign-aware Graph Transformer on the line graph. The entire model is unified and regularized by a Cross-View Co-training Mechanism for robust learning.

novel graph-native inductive biases, which are specifically engineered for the unique properties of signed networks.

**Line Graph Formulation.** We first transform the input signed bipartite graph, formally defined as $G_b = (\mathcal{U}, \mathcal{V}, \mathcal{E}, s)$, into its line graph dual $G_l = (\mathcal{V}_l, \mathcal{E}_l)$. Here, $s : \mathcal{E} \to \{-1, +1\}$ is the sign function. This transformation allows the model to directly reason about link-level interactions. The vertex set $\mathcal{V}_l$ corresponds to the edge set $\mathcal{E}$ of the original graph, such that each vertex $v_k \in \mathcal{V}_l$ represents a unique edge $e_k \in \mathcal{E}$. The edge set $\mathcal{E}_l$ is constructed based on edge incidence in $G_b$:

$$\mathcal{E}_l = \{(v_i, v_j) \mid v_i, v_j \in \mathcal{V}_l, i \neq j, \text{ and } e_i \cap e_j \neq \emptyset\}, \tag{1}$$

where an edge is treated as a set of its two endpoints. We provide a rigorous theoretical analysis in Appendix H, discussing the injectivity of this transformation based on Whitney's isomorphism theorem (guaranteed for bipartite graphs) and its computational complexity. In practice, the construction cost is linear $O(\Delta|\mathcal{E}|)$ due to the sparsity of real-world interaction graphs, and effectively converts higher-order signed motifs into one-hop neighborhoods for efficient attention learning.

Assuming initial node embeddings $\mathbf{H_U} \in \mathbb{R}^{|\mathcal{U}| \times d/2}$ and $\mathbf{H_V} \in \mathbb{R}^{|\mathcal{V}| \times d/2}$, we construct the initial feature matrix for the line graph, $\mathbf{X}_l \in \mathbb{R}^{|\mathcal{E}| \times d}$. The feature vector for a vertex $v_k$ representing edge $e_k = (u_p, v_q)$ is the concatenation of its endpoint embeddings:

$$\mathbf{x}_k = \mathbf{X}_l[k, :] = [\mathbf{H_U}[p, :] \,\|\, \mathbf{H_V}[q, :]]. \tag{2}$$

Finally, the labels for the line graph vertices are defined by a vector $\mathbf{Y}_l \in \{-1, +1\}^{|\mathcal{E}|}$, where $y_k = s(e_k)$. This formulation effectively reframes signed link prediction as a node classification task on the line graph.

**Graph-Native Inductive Priors.** Standard Transformers are inherently topology-agnostic, ignoring underlying graph structures. To address this, we inject two graph-native inductive priors directly into the self-attention mechanism. These priors provide multi-scale structural awareness, enabling the model to effectively reason over both global network-wide balance and local higher-order connectivity patterns.

**Relational Spectral Encoding (RSE)** serves as the global prior, designed to operationalize the principles of social balance theory within the spectral domain of the graph. To achieve this in a topologically sound manner, we first define the line graph's binary adjacency matrix, $\mathbf{A}_l \in \{0, 1\}^{|\mathcal{V}_l| \times |\mathcal{V}_l|}$, and a sign vector $\mathbf{s} \in \{-1, +1\}^{|\mathcal{V}_l|}$ where $s_k = \text{sign}(e_k)$. The topologically-aware signed adjacency matrix $\mathbf{A}_S$ is then constructed via the Hadamard product:

$$\mathbf{A}_S = \mathbf{A}_l \odot (\mathbf{s}\mathbf{s}^T). \tag{3}$$

The signed Laplacian is defined as $\mathbf{L}_S = \mathbf{D}_{|S|} - \mathbf{A}_S$, where $[\mathbf{D}_{|S|}]_{ii} = \sum_j |\mathbf{A}_S(i,j)|$. We construct the RSE prior from the eigenvectors corresponding to the smallest $d_h$ eigenvalues of $\mathbf{L}_S$:

$$\tilde{\mathbf{H}} = \text{EigVecs}_{\text{smallest } d_h}(\mathbf{L}_S), \qquad \mathbf{P}_s = \alpha_s \cdot \text{ZeroDiag}(\tilde{\mathbf{H}}\tilde{\mathbf{H}}^\top), \tag{4}$$

where the orthonormal columns of $\tilde{\mathbf{H}} \in \mathbb{R}^{|\mathcal{V}_l| \times d_h}$ encode global partitioning and balance akin to the Fiedler vector, enabling RSE to bypass local limits and capture long-range dependencies. Scaled by a learnable $\alpha_s \in \mathbb{R}$, we remove the diagonal to prevent self-attention dominance and treat $\mathbf{P}_s$ as fixed by stopping gradients to $\tilde{\mathbf{H}}$. The decomposition employs the Lanczos algorithm on $\mathbf{L}_S$, achieving a complexity of $\mathcal{O}(d_h \cdot K \cdot |\mathcal{E}_l|)$ for $K$ iterations.

**Topological Motif Encoding (TME)** complements the global prior by providing fine-grained local structural information. This component moves beyond simple adjacency to capture the semantic role of higher-order network motifs. Specifically, we focus on signed triadic closures, which manifest as paths of exactly two hops in the line graph. We define $N_p = 4$ distinct motif types based on the sign tuple of the edges forming a 2-hop path. This value is not an arbitrary hyperparameter but is naturally determined by the complete set of binary sign permutations for a 2-hop relation: $\{(+,+),(+,-),(-,+),(-,-)\}$, as empirically verified in Appendix K.

The TME prior, $\mathbf{P}_p$, is formulated:

$$\mathbf{o}_{ij} = \sum_{m \in \mathcal{S}_{ij}} \text{onehot}\big(s(e_{im}), s(e_{mj})\big) \in \mathbb{N}^4, \tag{5}$$

where $s(e_{im})$ is the sign of the edge between nodes $v_i$ and $v_m$. The final prior value is a learnable weighted sum of these counts, non-zero only for 2-hop neighbors:

$$\mathbf{P}_p(i,j) = \begin{cases} \alpha_p \cdot (\mathbf{o}_{ij}^\top \boldsymbol{\phi}) & \text{if } \text{dist}_l(i,j) = 2 \\ 0 & \text{otherwise,} \end{cases} \tag{6}$$

where $\boldsymbol{\phi} \in \mathbb{R}^4$ and $\alpha_p \in \mathbb{R}$ are learnable weights. This formulation allows the model to dynamically infer the importance of both the type and prevalence of local connectivity patterns. We keep $(+,-)$ and $(-,+)$ motifs distinct to respect the inherent directionality of the attention mechanism.

**Transformer Encoder Architecture.** The SGT encoder consists of a stack of $L$ identical layers, each layer being composed of two main sub-modules: multi-head self-attention (MHA) and a position-wise feed-forward network (FFN). We employ residual connections around each sub-module, followed by layer normalization.

The first sub-module, MHA, is where our graph-native inductive priors are injected. The input link representations $\mathbf{H}^{(l-1)} \in \mathbb{R}^{|\mathcal{V}_l| \times d}$ are first linearly projected into queries, keys, and values for each of the $N_h$ attention heads. For a given head $h$, the attention output is computed by augmenting the standard scaled dot-product attention with our structural priors:

$$\text{head}_h = \text{softmax}\left(\frac{(\mathbf{H}^{(l-1)}\mathbf{W}_h^Q)(\mathbf{H}^{(l-1)}\mathbf{W}_h^K)^T}{\sqrt{d_k}} + \mathbf{P}_s + \mathbf{P}_p\right)(\mathbf{H}^{(l-1)}\mathbf{W}_h^V), \tag{7}$$

where $\mathbf{W}_h^Q, \mathbf{W}_h^K, \mathbf{W}_h^V \in \mathbb{R}^{d \times d_k}$ are the learnable projection matrices for head $h$, and $d_k = d/N_h$. The outputs of all heads are then concatenated and passed through a final linear projection to produce the MHA output:

$$\text{MHA}(\mathbf{H}^{(l-1)}) = \text{Concat}(\text{head}_1, \ldots, \text{head}_{N_h})\mathbf{W}^O. \tag{8}$$

The full layer-wise update rule for transforming the input representations $\mathbf{H}^{(l-1)}$ to the output $\mathbf{H}^{(l)}$ at layer $l$ is defined as follows, where FFN is a two-layer perceptron applied to each position independently:

$$\mathbf{H}' = \text{LayerNorm}\left(\mathbf{H}^{(l-1)} + \text{MHA}(\mathbf{H}^{(l-1)})\right) \tag{9}$$

$$\mathbf{H}^{(l)} = \text{LayerNorm}\left(\mathbf{H}' + \text{FFN}(\mathbf{H}')\right). \tag{10}$$

The final output of the stack, $\mathbf{H}^{(L)}$, serves as the matrix of deeply contextualized and structurally-principled link representations used for downstream tasks.

## 4.2 HIERARCHICAL PROTOTYPICAL LEARNING OBJECTIVE

The structurally-principled representations $\mathbf{H}^{(L)}$ are optimized via a hierarchical objective. The theoretical premise of our approach is to reframe the learning problem as a probabilistic assignment over a set of learnable prototypes, a formulation designed to inherently address both class imbalance and intra-class multimodality. Crucially, these prototypes capture distinct, interpretable semantic patterns within the signed interaction data, as demonstrated in our empirical analysis in Appendix G We map each link embedding $\mathbf{h}_i \in \mathbf{H}^{(L)}$ to a probability distribution over a set of prototypes $\mathcal{C} = \bigcup_c \mathcal{C}_c$. The foundation of our objective is the soft assignment probability $p_{ij}$ of an embedding $\mathbf{h}_i$ to a prototype $\mathbf{c}_j$, governed by a softmax over the negative squared Euclidean distance $d(\mathbf{h}_i, \mathbf{c}_j) = \|\mathbf{h}_i - \mathbf{c}_j\|_2^2$:

$$p_{ij} = P(\mathbf{c}_j|\mathbf{h}_i) = \frac{\exp(-d(\mathbf{h}_i, \mathbf{c}_j)/\tau)}{\sum_{k=1}^{|\mathcal{C}|} \exp(-d(\mathbf{h}_i, \mathbf{c}_k)/\tau)}, \tag{11}$$

where $\tau$ is a temperature parameter. From this probabilistic foundation, we derive a composite loss $\mathcal{L}_{\mathrm{H}}$ with three synergistic components designed to sculpt the embedding space. The primary discriminative loss ($\mathcal{L}_{\mathrm{class}}$) applies a class-balanced cross-entropy to the marginalized class probability $P(y_i|\mathbf{h}_i) = \sum_{\mathbf{c}_k \in \mathcal{C}_{y_i}} p_{ik}$ to ensure accurate classification under imbalance:

$$\mathcal{L}_{\mathrm{class}} = -\sum_{i=1}^{N} \alpha_{y_i} \log P(y_i|\mathbf{h}_i), \tag{12}$$

where $\alpha_{y_i}$ is a class-balancing weight. Crucially, this loss operates on the marginal probability over multiple prototypes rather than a single centroid, allowing the model to capture diverse intra-class modes while $\alpha_{y_i}$ adjusts the decision boundary. This is complemented by a clustering regularizer ($\mathcal{L}_{\mathrm{cluster}}$), which minimizes the entropy of the assignment distribution $p_{ij}$ to enforce cluster compactness and encourage embeddings of the same class (including minority ones) to concentrate around specific prototypes:

$$\mathcal{L}_{\mathrm{cluster}} = \frac{1}{N} \sum_{i=1}^{N} \left( -\sum_{j=1}^{|\mathcal{C}|} p_{ij} \log p_{ij} \right). \tag{13}$$

Finally, a separation regularizer ($\mathcal{L}_{\mathrm{sep}}$) imposes a geometric prior on the prototypes themselves to ensure inter-class separation, which helps maintain large metric margins even when one class is heavily under-represented:

$$\mathcal{L}_{\mathrm{sep}} = \sum_{\mathbf{c}_k \in \mathcal{C}_{\mathrm{pos}}} \sum_{\mathbf{c}_j \in \mathcal{C}_{\mathrm{neg}}} \exp(-d(\mathbf{c}_k, \mathbf{c}_j)). \tag{14}$$

These components are jointly optimized in a weighted sum:

$$\mathcal{L}_{\mathrm{H}} = \mathcal{L}_{\mathrm{class}} + \beta_1 \mathcal{L}_{\mathrm{cluster}} + \beta_2 \mathcal{L}_{\mathrm{sep}}. \tag{15}$$

## 4.3 JOINT OPTIMIZATION WITH CROSS-VIEW CONSISTENCY

Training a deep encoder risks a *structure-semantics gap* where learned representations diverge from the topology. To mitigate this, we introduce a cross-view consistency mechanism that enforces topological fidelity by aligning two perspectives for each link $e_k$: the foundational structural view $\mathbf{h}_k^{(0)} \in \mathbf{X}_l$ and the advanced semantic view $\mathbf{h}_k^{(L)}$ from the SGT. Utilizing the learnable prototypes $\mathcal{C}$ as a shared latent vocabulary, we maximize the consistency of probabilistic assignments between these views. The probability of assigning an embedding $\mathbf{h}$ to a prototype $\mathbf{c}_j$ is given by:

$$P(\mathbf{c}_j|\mathbf{h}) = \frac{\exp(-\|\mathbf{h} - \mathbf{c}_j\|_2^2/\tau_c)}{\sum_{\mathbf{c}_m \in \mathcal{C}} \exp(-\|\mathbf{h} - \mathbf{c}_m\|_2^2/\tau_c)}, \tag{16}$$

where $\tau_c$ is a temperature parameter. For a given link $e_k$, this yields two probability distributions over the prototypes: $\mathbf{P}_k^{(0)}$ from the foundational view $\mathbf{h}_k^{(0)}$, and $\mathbf{P}_k^{(L)}$ from the advanced view $\mathbf{h}_k^{(L)}$. The cross-view consistency loss then penalizes the divergence between these two interpretations using the symmetric Kullback-Leibler (KL) divergence:

$$\mathcal{L}_{\mathrm{consistency}} = \frac{1}{2|\mathcal{V}_l|} \sum_{k=1}^{|\mathcal{V}_l|} \left( D_{\mathrm{KL}}(\mathbf{P}_k^{(0)}\|\mathbf{P}_k^{(L)}) + D_{\mathrm{KL}}(\mathbf{P}_k^{(L)}\|\mathbf{P}_k^{(0)}) \right). \tag{17}$$

**Overall Objective.** The entire HPC-SGT framework is then trained by jointly optimizing the hierarchical prototypical contrastive objective and the cross-view consistency loss. The final objective function is a weighted sum of these two components:

$$\mathcal{L}_{\text{total}} = \mathcal{L}_{\text{H}} + \gamma \mathcal{L}_{\text{consistency}}. \tag{18}$$

This joint optimization ensures that the SGT learns link representations that are not only discriminative and well-structured but also remain faithful to the ground-truth graph topology.

## 5 EXPERIMENTS

In this section, we empirically evaluate our proposed HPC-SGT framework. We first compare its performance against a wide range of state-of-the-art baselines on the task of signed link prediction. We then conduct detailed ablation studies to quantify the individual contributions of our core components. Finally, we analyze the framework's hyperparameter sensitivity and computational efficiency.

### 5.1 EXPERIMENTAL SETTINGS

**Datasets and Baselines.** We conduct experiments on four large-scale signed bipartite graph benchmarks: Amazon-Book McAuley et al. (2015), ML-1M Harper & Konstan (2015), ML-10M, and Gowalla Cho et al. (2011). We construct signed links following established protocols (Derr et al., 2018; Chen et al., 2024b). For rating-based datasets (Amazon-Book, MovieLens), we map user ratings $\geq 4$ to positive (+1) links and those $\leq 3$ to negative (-1). For the implicit check-in data from Gowalla, all existing interactions are considered positive, and we sample an equal number of unobserved user-location pairs as negative links. Detailed statistics for the resulting datasets are provided in Appendix A.1. We compare our framework, HPC-SGT, against fourteen state-of-the-art baselines spanning four categories: (i) Unsigned Methods, (ii) Early Signed Embeddings, (iii) GNN-based Models, and (iv) Transformer-based Architectures. Appendix A provides detailed statistics and descriptions.

**Evaluation Metrics.** Following established protocols (Zhang et al., 2023; Huang et al., 2021a), we evaluate performance on the signed link prediction task using four standard metrics: AUC, Binary-F1, Macro-F1, and Micro-F1. Higher values indicate superior performance. We use AUC as the primary metric for model selection, given its threshold-independent nature.

**Implementation Details and Protocol.** We adopt a standard transductive learning setup, splitting the links (nodes in the line graph) into 85% for training, 5% for validation, and 10% for testing. This standard link-based split, rather than a node-based one, ensures all node embeddings are learned during training while preventing label leakage, as the validation and test link instances are held out. For robustness, we report the mean performance over five independent runs with different random seeds. For a fair comparison, all models are initialized with identical 32-dimensional learnable node embeddings. Baselines operate on the original bipartite graph $G_b$, whereas our HPC-SGT operates on its line graph dual $G_l$. The final prediction score for a link $e_i$ is its probability of belonging to the positive class, $P(y_i = +1|\mathbf{h}_i)$, which is computed by marginalizing over the positive prototypes as defined in our methodology. All hyperparameters for all models were optimized via a systematic grid search, maximizing the AUC score on the validation set.

### 5.2 PERFORMANCE COMPARISON

Table 1 demonstrates that HPC-SGT establishes a new state-of-the-art across all benchmarks. By consistently surpassing GNNs like LightGCL, we validate the superiority of global attention over local message-passing for capturing long-range dependencies. Moreover, HPC-SGT outperforms Transformers like SIGformer, highlighting the decisive advantage of our graph-native inductive priors. Significant improvements in F1 metrics further underscore the efficacy of our hierarchical prototypical objective, which models multi-modal class structures and explicitly handles imbalance to yield robust decision boundaries compared to standard losses.

To ensure our gains stem from the integral design rather than artifacts of weighting strategies, we retrained baselines with identical class weights (Appendix E); HPC-SGT maintained a decisive lead

Table 1: Performance comparison on four signed bipartite datasets. Our proposed model, HPC-SGT, consistently outperforms all baseline methods across all metrics. The best results are highlighted in bold.

| Method | Amazon-Book | | | | ML-1M | | | | ML-10M | | | | Gowalla | | | |
|---|---|---|---|---|---|---|---|---|---|---|---|---|---|---|---|---|
| | AUC | Bi | Macro | Micro | AUC | Bi | Macro | Micro | AUC | Bi | Macro | Micro | AUC | Bi | Macro | Micro |
| DeepWalk | 0.594 | 0.573 | 0.538 | 0.585 | 0.591 | 0.557 | 0.523 | 0.539 | 0.627 | 0.580 | 0.542 | 0.579 | 0.574 | 0.502 | 0.511 | 0.548 |
| Node2Vec | 0.547 | 0.656 | 0.488 | 0.543 | 0.635 | 0.582 | 0.609 | 0.611 | 0.654 | 0.671 | 0.551 | 0.562 | 0.536 | 0.614 | 0.494 | 0.587 |
| LINE | 0.588 | 0.593 | 0.517 | 0.567 | 0.628 | 0.578 | 0.541 | 0.602 | 0.621 | 0.612 | 0.581 | 0.589 | 0.601 | 0.589 | 0.502 | 0.596 |
| SiNE | 0.594 | 0.559 | 0.491 | 0.458 | 0.573 | 0.528 | 0.511 | 0.579 | 0.609 | 0.534 | 0.528 | 0.499 | 0.585 | 0.526 | 0.545 | 0.602 |
| SBiNE | 0.578 | 0.541 | 0.486 | 0.509 | 0.552 | 0.503 | 0.509 | 0.516 | 0.582 | 0.522 | 0.519 | 0.586 | 0.596 | 0.528 | 0.508 | 0.557 |
| SCsc | 0.581 | 0.436 | 0.427 | 0.537 | 0.589 | 0.484 | 0.479 | 0.581 | 0.648 | 0.489 | 0.509 | 0.574 | 0.577 | 0.488 | 0.478 | 0.559 |
| SGCN | 0.593 | 0.693 | 0.504 | 0.582 | 0.632 | 0.662 | 0.615 | 0.627 | 0.632 | 0.671 | 0.584 | 0.605 | 0.602 | 0.651 | 0.518 | 0.604 |
| SGCL | 0.613 | 0.710 | 0.502 | 0.604 | 0.632 | 0.673 | 0.662 | 0.652 | 0.631 | 0.698 | 0.579 | 0.645 | 0.604 | 0.668 | 0.508 | 0.617 |
| SBGNN | 0.603 | 0.720 | 0.552 | 0.612 | 0.652 | 0.699 | 0.653 | 0.674 | 0.639 | 0.702 | 0.601 | 0.638 | 0.611 | 0.672 | 0.597 | 0.629 |
| SBGCL | 0.637 | 0.734 | 0.587 | 0.640 | 0.685 | 0.702 | 0.678 | 0.680 | 0.652 | 0.711 | 0.628 | 0.688 | 0.605 | 0.698 | 0.625 | 0.667 |
| LightGCL | 0.647 | 0.747 | 0.601 | 0.642 | 0.727 | 0.711 | 0.655 | 0.736 | 0.701 | 0.728 | 0.681 | 0.694 | 0.645 | 0.694 | 0.677 | 0.665 |
| SIGformer | 0.658 | 0.740 | 0.617 | 0.652 | 0.715 | 0.725 | 0.688 | 0.721 | 0.729 | 0.731 | 0.689 | 0.708 | 0.659 | 0.711 | 0.698 | 0.672 |
| SE-SGformer | 0.681 | 0.738 | 0.621 | 0.668 | 0.721 | 0.732 | 0.704 | 0.728 | 0.715 | 0.724 | 0.702 | 0.701 | 0.684 | 0.704 | 0.685 | 0.694 |
| **HPC-SGT (Ours)** | **0.744** | **0.801** | **0.671** | **0.718** | **0.748** | **0.781** | **0.734** | **0.745** | **0.760** | **0.784** | **0.735** | **0.747** | **0.739** | **0.753** | **0.721** | **0.736** |

despite marginal baseline improvements. We further validated the model's capacity for long-range dependencies via distance-bucket analysis (Appendix I), demonstrating superior stability on distant links where baselines degrade. Finally, comparisons against a "Line-GAT" baseline (Appendix J) confirm that the performance stems from our global sign-aware attention and inductive priors rather than solely from the line graph representation.

## 5.3 ABLATION STUDY

We evaluate four removals on Amazon-Book and ML-1M: the spectral prior (RSE), the motif prior (TME), the multi-prototype head (replaced by a single prototype per class), and the cross-view consistency term. All runs share the same training protocol and hyperparameters as the full model. Table 2 shows that across both datasets, each ablation yields a consistent drop on ranking and F1 metrics, while the full HPC-SGT remains strongest. The trends are complementary: RSE improves global structure awareness and class balance; TME benefits short-range decisions reflected in micro-averaged scores; the multi-prototype head better

Table 2: Ablation study of HPC-SGT's core components. Removing any module degrades performance, confirming its contribution. Best results are in bold.

| Method | Amazon-Book | | | | ML-1M | | | |
|---|---|---|---|---|---|---|---|---|
| | AUC | Bi | Ma | Mi | AUC | Bi | Ma | Mi |
| w/o RSE | 0.721 | 0.759 | 0.602 | 0.686 | 0.701 | 0.732 | 0.688 | 0.704 |
| w/o TME | 0.705 | 0.766 | 0.622 | 0.654 | 0.714 | 0.728 | 0.681 | 0.711 |
| Single-Prototype | 0.718 | 0.791 | 0.618 | 0.671 | 0.698 | 0.725 | 0.694 | 0.701 |
| w/o Cross-View | 0.701 | 0.784 | 0.630 | 0.689 | 0.717 | 0.719 | 0.684 | 0.723 |
| **HPC-SGT** | **0.744** | **0.801** | **0.671** | **0.718** | **0.739** | **0.753** | **0.721** | **0.736** |

captures intra-class variability than a single prototype; and the consistency term regularizes the encoder by aligning structural and semantic views. These effects appear in both benchmarks, indicating that the components address distinct failure modes rather than overlapping the same gain.

## 5.4 PARAMETER ANALYSIS

We examine four hyperparameters that map directly to the model design: the spectral capacity of the global prior ($d_h$), the number of prototypes per class ($K_c$), the cross-view consistency weight ($\gamma$), and the assignment-entropy weight ($\beta_1$). Each sweep varies a single parameter while holding the others fixed to the main configuration; unless swept, we set $\beta_1=0.2$ and $d_h=32$. The curves (AUC and Bi-F1) show a consistent pattern. Increasing $d_h$ improves performance up to a clear knee and then saturates, indicating that a modest set of spectral components is sufficient to carry the long-range balance signal. Varying $K_c$ confirms the utility of explicit multi-prototype modeling: moving from one to a small set of prototypes strengthens decision boundaries, after which gains diminish as the head becomes over-parameterized. The consistency term exhibits a broad plateau, with moderate $\gamma$ yielding the best trade-off between anchoring the encoder to the structural view and preserving flexibility. For $\beta_1$, mid-range values avoid both diffuse assignments and premature peaking, and produce more stable training.

## 5.5 CLASS IMBALANCE.

To evaluate the performance of HPC-SGT in addressing extreme class imbalance issues, we select the Bonanza dataset for our experiments, which exhibits the largest disparity between the number of positive and negative links. We compare HPC-SGT with several existing methods that address class imbalance among graph nodes, including ImGAGN, GraphENS, and GraphSHA. To process the signed bipartite graph, we transform it into a line graph before applying the aforementioned methods. The experimental results are shown in Figure 4. It is observed that HCP-SGT significantly outperforms ImGAGN Wang et al. (2024), GraphENS Shi et al.

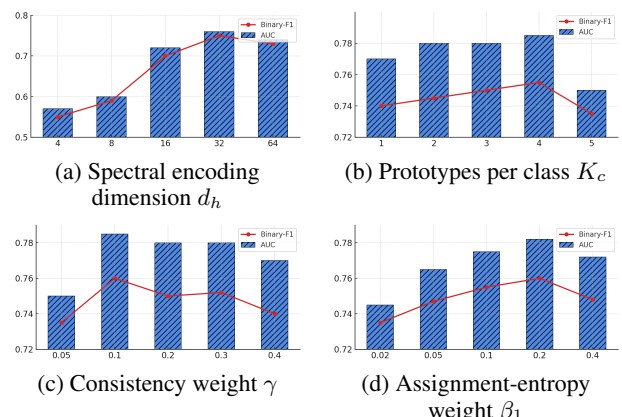

(a) Spectral encoding dimension $d_h$

(b) Prototypes per class $K_c$

(c) Consistency weight $\gamma$

(d) Assignment-entropy weight $\beta_1$

Figure 3: Sensitivity analysis of key hyper-parameters on model performance.

(2024), and GraphSHA. This demonstrates that HPC-SGT is better at balancing performance among different classes, particularly excelling in the precise identification of the tail class.

To further rigorously validate this capability, we conducted two additional sets of experiments detailed in Appendix F: (1) benchmarking against strong general baselines (LightGCL, SIGformer, SE-SGformer) on the highly skewed Bonanza dataset, and (2) performing stress tests on the Amazon-Book dataset with artificially induced extreme imbalance ratios (90:10 and 95:5). In both scenarios, HPC-SGT maintains a robust performance advantage, particularly in Macro-F1 scores, confirming its resilience to severe data skew.

## 6 CONCLUSION

In this paper, we introduced HPC-SGT, a novel framework for signed link prediction. Extensive experiments and ablation studies validated that HPC-SGT establishes a new state-of-the-art on multiple benchmark datasets. We demonstrated that its success stems from a principled co-design of its components. Its Sign-aware Graph Transformer operates on the line graph with novel inductive priors to capture global dependencies inaccessible to standard GNNs. This powerful encoder is guided by a hierarchical prototypical objective that synergistically handles complex data, using a class-balanced loss to

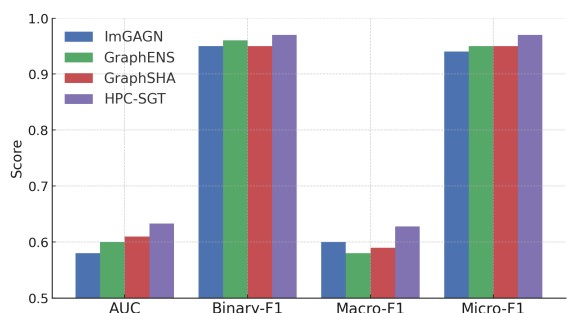

Figure 4: Comparison on BONANZA under severe class imbalance.

manage imbalance and geometric regularizers to model multimodality. The framework's robustness is further enhanced by a cross-view consistency mechanism that ensures topological fidelity. Acknowledging the computational cost of the Transformer as a limitation, a key direction for future work is the exploration of more efficient sparse attention mechanisms. Furthermore, while this work focuses on static snapshots, HPC-SGT is naturally extensible to dynamic settings. Future research could deploy the SGT encoder over temporal sliding windows and maintain hierarchical prototypes via incremental updates to effectively model evolving signed interactions.

ETHICS STATEMENT

The authors of this work have adhered to the ICLR Code of Ethics. Our research is conducted on publicly available benchmark datasets commonly used for evaluating signed link prediction models. These datasets contain anonymized user-item interactions, and our study does not involve any direct experimentation with human subjects.

We acknowledge that link sign prediction models, including HPC-SGT, have potential for dual use. While they can be applied beneficially to enhance recommendation systems or identify supportive communities, they could also be misused to infer contentious social relationships or amplify polarization. Furthermore, as our model is trained on real-world data, it may inherit and potentially amplify existing societal biases present in that data. A thorough investigation into the fairness and potential biases of the learned representations is an important direction for future research. We are committed to the responsible development and application of machine learning technologies.

REPRODUCIBILITY STATEMENT

We are committed to ensuring the reproducibility of our research. To this end, we provide the main source code for our HPC-SGT framework, model configurations, and experiment scripts in the supplementary materials. The core methodology, including the architecture of our Sign-aware Graph Transformer and the formulation of our hierarchical learning objective, is detailed in Section 4. A step-by-step training procedure is provided in pseudocode in Appendix B. All experimental settings, including descriptions of the publicly available benchmark datasets, evaluation protocols, and a comprehensive list of hyperparameter values, are documented in Section 5 and the Appendix.

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



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

# A DATASETS AND EXPERIMENTAL SETTINGS

## A.1 DATASETS

Table 3 lists the basic statistics for every signed bipartite graph used in our study. The four core benchmarks analysed in the main paper—*Amazon-Book*, *ML-10M*, *Gowalla*, and *MovieLens-1M*—cover e-commerce, location-based social check-ins, and movie-rating domains. To gauge scalability and robustness, we further consider five supplementary signed graphs of varying size and sparsity: *Bonanza* (e-commerce buyer/seller ratings), *U.S. House* and *U.S. Senate* (congressional roll-call votes), and two phases of a computer-science *Review* dataset (pre- and post-rebuttal).

Table 3: Statistics of the signed bipartite graphs used in this work. $|U|$ / $|V|$ denote the two node partitions; $|E| = |E^+|+|E^-|$. Positive/negative ratios follow the sign definitions in each source.

| Dataset | $|U|$ | $|V|$ | $|E|$ | $\%|E^+|$ | $\%|E^-|$ | Domain |
|---|---|---|---|---|---|---|
| Amazon-Book | 35,736 | 38,121 | 1,960,674 | 0.806 | 0.194 | E-commerce |
| ML-10M | 69,878 | 10,677 | 10,000,054 | 0.589 | 0.411 | Movies |
| Gowalla | 37,000 | 11,500 | 3,500,000$^{\dagger}$ | 0.612 | 0.388 | LBSN |
| MovieLens-1M | 6,040 | 3,952 | 1,000,209 | 0.575 | 0.425 | Movies |
| Bonanza | 7,919 | 1,973 | 36,543 | 0.980 | 0.020 | E-commerce |
| U.S. House | 515 | 1,281 | 114,378 | 0.540 | 0.460 | Politics |
| U.S. Senate | 145 | 1,056 | 27,083 | 0.553 | 0.447 | Politics |
| Review (Pre.) | 182 | 304 | 1,170 | 0.403 | 0.597 | Peer review |
| Review (Final) | 182 | 304 | 1,170 | 0.397 | 0.603 | Peer review |

## A.2 BASELINE DESCRIPTIONS

To provide a comprehensive comparison, we evaluate HPC-SGT against four families of existing techniques. All baselines are trained and tuned under the unified protocol described in the paper; hyper-parameter grids follow the ranges recommended by the original authors.

**(i) Heuristic / unsigned methods.** DeepWalk, Node2Vec, and LINE learn node embeddings from unsigned random walks or edge sampling, entirely disregarding polarity. Once the embeddings are obtained, the representation of a candidate edge is formed by concatenating the two endpoint vectors and feeding the result into a logistic classifier. Although these models cannot reason about positive versus negative semantics, they establish a structural lower bound and clarify how much benefit arises purely from sign information.

**(ii) Early signed embeddings.** SiNE, SBiNE, and SCsc extend skip-gram training with sign-aware constraints. SiNE introduces a margin-based triplet loss that forces positively connected nodes to be closer than negatively connected ones. SBiNE tailors this idea to bipartite topology by preserving sign-specific first- and second-order proximity. SCsc adds social-balance regularisers that push the geometry of embeddings toward structurally balanced configurations. All three methods remain shallow and scalable, yet their expressiveness is limited by the absence of higher-order message passing.

**(iii) GNN-based models.** SGCN propagates messages through disjoint positive and negative channels, explicitly following balance theory at each layer. SGCL introduces a contrastive loss that pulls together node pairs appearing in balanced triads and pushes apart those in unbalanced ones. SBGNN augments standard graph convolutions with learnable sign masks, while SBGCL combines this architecture with contrastive regularisation to sharpen sign discrimination. LightGCL streamlines the same principles into a parameter-efficient design that reduces memory without sacrificing accuracy. These models capture high-order structures but still aggregate over the entire graph, leaving local sign motifs partly diluted and node roles implicit.

**(iv) Transformer-style architectures.** SIGformer applies multi-head self-attention to signed graphs, masking attention weights with polarity-aware filters to preserve balance constraints even

at long range. SE-SGformer extends this blueprint with self-explainable heads that highlight path patterns responsible for each prediction. Both architectures rely on global attention, which can blur local context and incurs quadratic memory growth with graph size.

### A.3 Implementation Details

**Experimental Setup.** All experiments were conducted on a server equipped with an NVIDIA A100 GPU. Our proposed HPC-SGT framework was implemented using PyTorch and the PyTorch Geometric (PyG) library. For all baseline models, we utilized their officially released code where available or re-implemented them faithfully according to their original papers to ensure a rigorous and fair comparison.

**HPC-SGT Configuration and Training.** Unless otherwise specified, we set the embedding dimension to $d = 32$ for all models. For our HPC-SGT, the Sign-aware Graph Transformer encoder consists of $L = 3$ layers, with $N_h = 4$ attention heads in each MHA module. The hidden dimension of the FFN was set to 256. We applied a dropout rate of 0.1 throughout the encoder. For our graph-native inductive priors, the spectral dimension for RSE was set to $d_h = 32$.

The hierarchical prototypical objective is configured with $K_c = 4$ prototypes per class and a temperature of $\tau = 0.1$ for the probabilistic assignment. The loss weights were set to $\beta_1 = 0.2$ for the clustering regularizer and $\beta_2 = 0.1$ for the separation regularizer. For the cross-view consistency mechanism, the temperature was set to $\tau_c = 1.0$ and the loss weight to $\gamma = 0.1$. All models were trained using the AdamW optimizer with a learning rate of $1 \times 10^{-3}$ and a weight decay of $1 \times 10^{-5}$, managed by a cosine annealing scheduler. We used a batch size of 1024 and trained for up to 200 epochs, with an early stopping patience of 20 epochs based on the validation set's AUC score.

## B Algorithm Details

Algorithm 1 provides a detailed outline of the training procedure for our proposed HPC-SGT framework. The process is divided into two main stages: initialization and the main training loop, where all components are jointly optimized.

**Stage 1: Initialization (Lines 3-6).** Before training, we first construct the line graph dual $G_l$ from the input signed bipartite graph $G_b$. The initial node embeddings for the original graph, $\mathbf{H}_U$ and $\mathbf{H}_V$, are initialized as learnable parameters. These are then used to construct the initial feature matrix for the line graph, $\mathbf{H}^{(0)}$, by concatenating the endpoint embeddings for each corresponding link. Finally, the parameters of the SGT Encoder and the set of learnable prototypes $\mathcal{C}$ are initialized.

**Stage 2: Joint Optimization Loop (Lines 8-16).** The main training is performed by jointly optimizing all learnable parameters within a training loop. In each epoch, the SGT Encoder first processes the initial link features $\mathbf{H}^{(0)}$ to produce the final, contextualized link embeddings $\mathbf{H}^{(L)}$ (Line 10). These embeddings are then used to compute the two main components of our total loss function.

- The Hierarchical Prototypical Objective, $\mathcal{L}_{\mathrm{H}}$, is computed based on the final embeddings $\mathbf{H}^{(L)}$ and the prototypes $\mathcal{C}$. This involves first calculating the probabilistic assignments of embeddings to prototypes, and then using these probabilities to compute the three synergistic loss terms: the class-balanced discriminative loss ($\mathcal{L}_{\mathrm{class}}$), the clustering regularizer ($\mathcal{L}_{\mathrm{cluster}}$), and the separation regularizer ($\mathcal{L}_{\mathrm{sep}}$) (Line 12).

- The Cross-View Consistency Loss, $\mathcal{L}_{\mathrm{consistency}}$, is computed by measuring the distributional divergence between the prototype assignments derived from the initial (structural) link features $\mathbf{H}^{(0)}$ and the final (semantic) link embeddings $\mathbf{H}^{(L)}$ (Line 14).

Finally, these two objectives are combined into a single total loss $\mathcal{L}_{\mathrm{total}}$, and the gradient is back-propagated to jointly update all learnable parameters of the framework: the SGT, the prototypes $\mathcal{C}$, and the initial node embeddings $\mathbf{H}_U$ and $\mathbf{H}_V$ (Line 16).

---

**Algorithm 1** HPC-SGT Training Procedure

---

1: **Input:** Signed bipartite graph $G_b = (\mathcal{U}, \mathcal{V}, \mathcal{E}, s)$, hyperparameters $\beta_1, \beta_2, \gamma, \tau, \ldots$.
2: **Output:** Trained SGT Encoder, Prototypes $\mathcal{C}$, and Node Embeddings $\mathbf{H}_U, \mathbf{H}_V$.
   *// — Stage 1: Initialization —*
3: Initialize learnable node embeddings $\mathbf{H}_U, \mathbf{H}_V$.
4: Construct line graph $G_l = (\mathcal{V}_l, \mathcal{E}_l)$ from $G_b$.
5: Construct initial link features $\mathbf{H}^{(0)}$ where $\mathbf{h}_k^{(0)} = [\mathbf{H}_U[p,:] \, \| \, \mathbf{H}_V[q,:]]$ for link $e_k = (u_p, v_q)$.
6: Initialize parameters for SGT Encoder and Prototypes $\mathcal{C}$.
   *// — Stage 2: Joint Optimization Loop —*
7: **for** each training epoch **do**
8:    *// Forward pass to get final link embeddings*
9:    $\mathbf{H}^{(L)} = \text{SGT\_Encoder}(\mathbf{H}^{(0)}, G_l)$
10:   *// Compute Hierarchical Prototypical Objective*
11:   Compute $\mathcal{L}_{\text{H}} = \mathcal{L}_{\text{class}} + \beta_1 \mathcal{L}_{\text{cluster}} + \beta_2 \mathcal{L}_{\text{sep}}$ using $\mathbf{H}^{(L)}$ and $\mathcal{C}$.
12:   *// Compute Consistency Loss*
13:   Compute $\mathcal{L}_{\text{consistency}}$ between assignments from $\mathbf{H}^{(0)}$ and $\mathbf{H}^{(L)}$.
14:   *// Combine objectives and update all parameters*
15:   $\mathcal{L}_{\text{total}} = \mathcal{L}_{\text{H}} + \gamma \mathcal{L}_{\text{consistency}}$
16:   Update all parameters ($\mathbf{H}_U, \mathbf{H}_V$, SGT, $\mathcal{C}$) via backpropagation.
17: **end for**
18: **return** Trained parameters.

---

# C  ADDITIONAL RESULTS ON BENCHMARK DATASETS

We evaluate our approach on four additional benchmark datasets; the full results are reported in Table 4. As a first observation, network–embedding techniques markedly improve signed link prediction: unsigned methods such as DeepWalk, Node2Vec, and LINE already outperform random embeddings even though they ignore edge polarity. Against this backdrop, HPC-SGT delivers the strongest performance on nearly every metric and dataset. The improvement is particularly pronounced on *Bonanza*, where HPC-SGT raises the Macro-F1 score by over 10% relative to the strongest baseline without reducing Micro-F1, indicating that the model boosts recall on minority (tail) classes while preserving accuracy on majority (head) classes.

Table 4: Performance comparison on four additional benchmark datasets: *U.S. House*, *U.S. Senate*, *Review*, and *Bonanza*. For all metrics, higher is better. Our method, HPC-SGT, demonstrates consistently superior performance.

| | U.S. House | | | | U.S. Senate | | | | Review | | | | Bonanza | | | |
|---|---|---|---|---|---|---|---|---|---|---|---|---|---|---|---|---|
| Method | AUC | Binary | Macro | Micro | AUC | Binary | Macro | Micro | AUC | Binary | Macro | Micro | AUC | Binary | Macro | Micro |
| Random | 0.541 | 0.560 | 0.540 | 0.541 | 0.543 | 0.568 | 0.542 | 0.543 | 0.556 | 0.510 | 0.553 | 0.556 | 0.529 | 0.735 | 0.389 | 0.590 |
| Deepwalk | 0.615 | 0.636 | 0.614 | 0.615 | 0.623 | 0.653 | 0.622 | 0.623 | 0.625 | 0.580 | 0.620 | 0.625 | 0.629 | 0.791 | 0.433 | 0.660 |
| Node2Vec | 0.633 | 0.651 | 0.632 | 0.633 | 0.645 | 0.670 | 0.644 | 0.645 | 0.653 | 0.620 | 0.645 | 0.649 | 0.626 | 0.759 | 0.416 | 0.619 |
| LINE | 0.580 | 0.611 | 0.579 | 0.580 | 0.569 | 0.611 | 0.568 | 0.569 | 0.620 | 0.593 | 0.607 | 0.610 | 0.617 | 0.702 | 0.382 | 0.545 |
| SiNE | 0.611 | 0.623 | 0.610 | 0.611 | 0.590 | 0.599 | 0.589 | 0.590 | 0.620 | 0.959 | 0.559 | 0.931 | 0.582 | 0.533 | 0.575 | 0.582 |
| SBiNE | 0.835 | 0.843 | 0.834 | 0.835 | 0.811 | 0.825 | 0.810 | 0.811 | 0.549 | 0.424 | 0.530 | 0.557 | 0.561 | 0.857 | 0.460 | 0.753 |
| SCsc | 0.827 | 0.837 | 0.826 | 0.827 | 0.816 | 0.829 | 0.814 | 0.816 | 0.552 | 0.336 | 0.482 | 0.581 | 0.652 | 0.643 | 0.354 | 0.484 |
| MFwBT | 0.809 | 0.823 | 0.809 | 0.810 | 0.785 | 0.804 | 0.785 | 0.786 | 0.472 | 0.434 | 0.469 | 0.475 | 0.577 | 0.892 | 0.481 | 0.807 |
| SBRW | 0.822 | 0.833 | 0.821 | 0.822 | 0.814 | 0.829 | 0.813 | 0.814 | 0.583 | 0.542 | 0.576 | 0.581 | 0.531 | 0.982 | 0.535 | 0.965 |
| SGCN | 0.808 | 0.827 | 0.808 | 0.809 | 0.815 | 0.827 | 0.815 | 0.816 | 0.610 | 0.593 | 0.601 | 0.637 | 0.587 | 0.896 | 0.487 | 0.814 |
| SGCL | 0.824 | 0.835 | 0.824 | 0.824 | 0.820 | 0.834 | 0.820 | 0.820 | 0.729 | 0.656 | 0.631 | 0.633 | 0.584 | 0.987 | 0.514 | 0.974 |
| SBGNN | 0.848 | 0.856 | 0.847 | 0.847 | 0.824 | 0.832 | 0.821 | 0.822 | 0.674 | 0.636 | 0.662 | 0.666 | 0.576 | 0.961 | 0.540 | 0.926 |
| SBGCL | 0.810 | 0.811 | 0.807 | 0.807 | 0.809 | 0.818 | 0.808 | 0.809 | 0.748 | 0.706 | 0.747 | 0.754 | 0.590 | 0.973 | 0.558 | 0.947 |
| HPC-SGT | 0.871 | 0.894 | 0.870 | 0.871 | 0.853 | 0.869 | 0.853 | 0.853 | 0.800 | 0.767 | 0.799 | 0.803 | 0.623 | 0.989 | 0.616 | 0.979 |

# D  COMPUTATIONAL EFFICIENCY

## D.1  COMPUTATIONAL EFFICIENCY COMPARISON

To evaluate the computational efficiency of different methods, we conducted comparative experiments on four datasets of varying sizes. We measured the training time required per epoch in seconds, with the results recorded in Table 5. The findings reveal a clear trade-off between model

Table 5: Comparison with various methods concerning time consumption.

|         | Review | Bonanza | U.S. House | U.S. Senate |
|---------|--------|---------|------------|-------------|
| SBGNN   | **0.106** | 0.881   | 0.527      | **0.312**   |
| SBGCL   | 0.703  | 1.096   | 0.994      | 0.856       |
| HPC-SGT | 0.439  | **0.575** | **0.491**  | 0.402       |

complexity and performance. On smaller-scale datasets like Review, simpler GNN-based methods demonstrate lower time consumption due to their lightweight architecture. Conversely, on graph datasets with a larger number of links, such as Bonanza, the runtime of our HPC-SGT becomes more comparable to other state-of-the-art models. While running a Transformer on the line graph is inherently more costly than shallow GNNs, the sparsity of real-world signed interactions ensures that the line graph scale remains controllable, avoiding theoretical worst-case density. We argue that this moderate computational overhead is a reasonable trade-off to achieve the concrete performance gains documented in our experiments, particularly the substantial improvements in Macro-F1 and minority class recall under severe class imbalance (as detailed in Appendix F).

## D.2 Scalability Stress Test and Component Decomposition

To rigorously assess the scalability of HPC-SGT with respect to line graph size, of different sizes from the Amazon-Book dataset, where the number of links $|\mathcal{E}|$ corresponds to the number of nodes in the line graph. We measured the training time per epoch and peak GPU memory usage on an NVIDIA A100 (80GB). The results, presented in Table 6, are consistent with the expected behavior of a full self-attention layer on the line graph: memory and time grow noticeably as the number of links approaches 100k, but remain practical in the regime we actually operate in. This is compatible with the runtime we report in Table 5 and supports our claim that HPC-SGT is feasible for medium-to-large signed bipartite graphs.

Table 6: Scalability profile of HPC-SGT on sampled Amazon-Book subgraphs.

| # Links (Line-Graph Nodes) | 10k | 20k | 40k | 80k | 100k |
|-----------------------------|------|------|------|------|------|
| Time (s/epoch)              | 0.06 | 0.16 | 0.45 | 1.80 | 2.95 |
| Peak Memory (GB)            | 0.7  | 1.5  | 5.2  | 18.5 | 28.4 |

To identify the primary bottleneck, we also profiled memory usage by component on a batch with 50k links. The distribution is as follows:

- **Attention Matrix** (Line-graph self-attention): $\approx 82\%$ of total GPU memory.
- **Graph Structure** (Line-graph adjacency & RSE priors): $\approx 11\%$.
- **Parameters** (Embeddings, gradients, and prototypes): $\approx 7\%$.

This decomposition confirms that the full attention matrix is the dominant cost factor (82%), while our prop/osed inductive priors (RSE/TME) and hierarchical prototype heads add only moderate overhead. These findings validate our feasibility claims for sparse, real-world signed graphs and directly motivate future work on incorporating sparse attention mechanisms to reduce the dominant term toward linear scaling.

## E Fairness Comparison with Class-Weighted Baselines

To ensure a fair comparison regarding the handling of class imbalance, we retrained the strongest baseline models (LightGCL, SIGformer, SE-SGformer) using the same class-weighted loss formulation as our HPC-SGT. Specifically, we replaced their standard losses with the weighted objective defined in Eq. (12), balancing the contribution of positive and negative samples based on training set statistics. Table 7 reports the comparison results on Amazon-Book and ML-1M. The results show that while re-weighting provides a modest performance boost to the baselines, HPC-SGT still

Table 7: Performance comparison with baselines retrained using the same class-weighted loss as HPC-SGT.

| Method | Amazon-Book | | ML-1M | |
|---|---|---|---|---|
| | AUC | Binary-F1 | AUC | Binary-F1 |
| LightGCL | 0.647 | 0.747 | 0.727 | 0.711 |
| LightGCL (w/ class weights) | 0.652 | 0.755 | 0.731 | 0.720 |
| SIGformer | 0.658 | 0.740 | 0.715 | 0.725 |
| SIGformer (w/ class weights) | 0.663 | 0.748 | 0.719 | 0.733 |
| SE-SGformer | 0.681 | 0.738 | 0.721 | 0.732 |
| SE-SGformer (w/ class weights) | 0.688 | 0.748 | 0.727 | 0.742 |
| **HPC-SGT (Ours)** | **0.744** | **0.801** | **0.748** | **0.781** |

consistently outperforms them across all metrics. This validates that the superiority of our framework is driven by the Sign-aware Graph Transformer architecture and the hierarchical prototypical objective, rather than simply by the application of class weights.

## F    EXTENDED ANALYSIS ON CLASS IMBALANCE

We provide a comprehensive evaluation of HPC-SGT under severe class imbalance through two additional analyses.

**Comparison with General Baselines on Bonanza.**   In the main text, we compared HPC-SGT with imbalance-specific methods on the Bonanza dataset. Here, we extend this comparison to the strongest general baselines identified in Table 1: LightGCL, SIGformer, and SE-SGformer. As shown in Table 8, HPC-SGT outperforms these strong baselines, achieving the highest AUC and, notably, a significantly higher Macro-F1. This indicates that our hierarchical prototypical objective effectively prevents minority classes from being submerged, a common failure mode for general models in such extreme settings.

Table 8: Performance comparison with strong general baselines on the extremely imbalanced Bonanza dataset.

| Method | Bonanza AUC | Bonanza Macro-F1 |
|---|---|---|
| LightGCL | 0.584 | 0.521 |
| SIGformer | 0.599 | 0.543 |
| SE-SGformer | 0.603 | 0.574 |
| **HPC-SGT (Ours)** | **0.623** | **0.616** |

**Stress-Testing on Amazon-Book.**   To assess robustness on standard benchmarks, we constructed artificially skewed versions of the Amazon-Book dataset by down-sampling negative links to create training sets with 90:10 and 95:5 positive-to-negative ratios. We then re-trained and evaluated all models. The results in Table 9 show that while performance naturally degrades with increased imbalance, HPC-SGT maintains a clear lead. In the most severe 95:5 scenario, our advantage in Macro-F1 becomes even more prominent, mutually confirming the findings from the Bonanza dataset.

## G    PROTOTYPE INTERPRETABILITY ANALYSIS

To explore the semantic meaning of the learned prototypes, we conducted a quantitative analysis on the Amazon-Book dataset. For each learned prototype, we retrieved the top-100 links with the highest assignment probability $p_{ij}$. We then calculated the mean rating (from original 1-5 star data),

Table 9: Stress-test results on Amazon-Book with artificially induced extreme imbalance ratios (90:10 and 95:5).

| Method | A-Book (90:10) | | A-Book (95:5) | |
|---|---|---|---|---|
| | AUC | Macro-F1 | AUC | Macro-F1 |
| LightGCL | 0.641 | 0.571 | 0.631 | 0.514 |
| SIGformer | 0.652 | 0.585 | 0.642 | 0.529 |
| SE-SGformer | 0.672 | 0.593 | 0.664 | 0.561 |
| **HPC-SGT (Ours)** | **0.736** | **0.645** | **0.731** | **0.627** |

Table 10: Semantic interpretation of learned prototypes on Amazon-Book. Statistics are computed from the top-100 links assigned to each prototype.

| Proto | Class | Mean Rating | % Negative | Balanced Ratio | Interpretation |
|---|---|---|---|---|---|
| P1 | Positive | 4.85 | 4% | 0.85 | High-Confidence Positives |
| P2 | Positive | 4.68 | 8% | 0.77 | Community Favorites |
| P3 | Positive | 4.45 | 15% | 0.68 | Noisy/Weak Positives |
| N1 | Negative | 1.45 | 90% | 0.70 | Strong Rejection (1-star) |
| N2 | Negative | 2.15 | 82% | 0.58 | Disappointment (Mixed 1–2) |
| N3 | Negative | 2.95 | 70% | 0.48 | Borderline (Mostly 3-star) |

the proportion of negative links, and the ratio of links involved in structurally balanced triads (based on TME statistics). The results for representative prototypes are summarized in Table 10.

From these statistics, we can see that the prototype is not a random cluster, but corresponds to an intuitive user item interaction mode. P1 and P2 obviously correspond to "very strong positive feedback": the average score is very high, there are few negative links, and the structure is highly balanced; P3 tends to be "noise/weak positive", with a higher proportion of negative links and a lower structural balance. In the negative category, N1 corresponds to a strong rejection dominated by 1 star, N2 reflects the disappointed behavior of swinging between 1 and 2 stars, and N3 focuses on the critical samples dominated by 3 stars, and most of them appear in areas with less balanced structure. In this way, the regular geometric picture of multi prototype distribution and separation can be directly mapped to the semantic and structural patterns in the actual data, so as to enhance the interpretability of hierarchical prototype targets.

# H  THEORETICAL ANALYSIS OF LINE GRAPH TRANSFORMATION

In this section, we address the theoretical soundness of the line graph transformation regarding isomorphism and computational complexity.

## H.1  INJECTIVITY AND WHITNEY'S ISOMORPHISM THEOREM

A key theoretical concern is whether the line graph transformation is lossless (i.e., whether $G$ can be recovered from $L(G)$). Whitney's Isomorphism Theorem states that for connected graphs with more than 3 vertices, if $L(G) \cong L(G')$, then $G \cong G'$. The only classical exceptions are the triangle $K_3$ and the claw $K_{1,3}$, which share the same line graph ($K_3$). However, in our setting, the input $G$ is strictly a bipartite graph. Since bipartite graphs cannot contain odd cycles, the triangle $K_3$ is structurally impossible. Thus, the ambiguity in Whitney's theorem is explicitly excluded, ensuring that the transformation preserves the topological structure uniquely.

Furthermore, our method does not rely solely on the line graph being a perfect topological inverse. Each node $v_k$ in the line graph (representing edge $e_k = (u, v)$ in $G_b$) is enriched with content-related features as defined in Eq. (11)

$$\mathbf{x}_k = [\mathbf{H}_U[u] \, \| \, \mathbf{H}_V[v] \, \| \, \text{sign}(e_k) \, \| \, \text{RSE/TME stats}].$$

Even if two bipartite graphs were to map to the same line graph topology, their differences would be preserved in these node attributes (endpoint embeddings and signs), ensuring the model captures the distinct identity of the original graph.

## H.2 COMPLEXITY ANALYSIS

Let $G = (U, V, E, s)$ be a signed bipartite graph with degrees $\{d_x\}_{x \in U \cup V}$. The line graph $L(G)$ has $|V_\ell| = |E|$ nodes. The number of edges in the line graph, $|E_\ell|$, is determined by the shared endpoints in the original graph:

$$|E_\ell| = \sum_{u \in U} \binom{d_u}{2} + \sum_{v \in V} \binom{d_v}{2}.$$

The construction process involves enumerating unordered pairs of incident edges for each node, with a time complexity of $O(\sum_x d_x^2)$. In the worst case, this is $O(\Delta|E|)$, where $\Delta$ is the maximum degree. However, real-world recommendation and interaction graphs are typically sparse, where $\Delta \ll |E|$. Consequently, the construction complexity is linear in practice. As shown in our efficiency analysis (Table 5 and Appendix D.2), the preprocessing cost is negligible compared to the Transformer layers.

As shown in our efficiency analysis (Table 5 and Appendix D.2), the preprocessing cost is negligible compared to the Transformer layers.

**Sparse Implementation of Signed Adjacency (Eq. 3).** We explicitly clarify the implementation of the signed structural matrix $\mathbf{A}_S = \mathbf{A}_l \odot (\mathbf{s}\mathbf{s}^T)$ defined in Eq. (3). The outer product notation $\mathbf{s}\mathbf{s}^T$ is used solely to express the mathematical logic of sign interactions. In our implementation, we strictly avoid materializing a dense $|\mathcal{V}_l| \times |\mathcal{V}_l|$ matrix. Instead, we exploit the sparsity of the line graph adjacency $\mathbf{A}_l$ and compute the sign product $s_i s_j$ only for non-zero entries:

$$\mathbf{A}_S(i, j) = \mathbf{A}_l(i, j) \cdot s_i s_j, \quad \forall (i, j) \text{ s.t. } \mathbf{A}_l(i, j) \neq 0.$$

This element-wise scaling operates strictly on the non-zero values of the sparse tensor (CSR/COO) format). Consequently, the time and memory complexity is $O(|\mathcal{E}_l|)$, where $|\mathcal{E}_l|$ is the number of edges in the line graph. For sparse recommender-style graphs, $|\mathcal{E}_l| \ll |\mathcal{V}_l|^2$, rendering this step effectively linear and computationally negligible compared to the subsequent Transformer layers.

Finally, this transformation serves a critical functional role: it converts "two-hop paths" and signed triangles in the original graph into one-hop neighborhoods in the line graph. This allows our Topological Motif Encoding (TME) to capture signed triangular motifs directly within a single attention layer, avoiding the need for deep stacks of message-passing layers to approximate these higher-order structures.

## I LONG-RANGE DEPENDENCY ANALYSIS

To empirically quantify the model's ability to capture long-range dependencies, we performed a distance-bucket analysis on the Amazon-Book dataset. We constructed a user-user projection graph based on co-interactions and computed the shortest-path distance for each test link. We then grouped test links into three buckets: Short-range ($\leq$ 3-hop), Mid-range (5-hop), and Long-range ($\geq$ 7-hop).

We compared HPC-SGT against a representative GNN (LightGCL) and a strong Transformer baseline (SE-SGformer). The AUC performance per bucket is reported in Table 11.

Table 11: Distance-bucket analysis on Amazon-Book. We report AUC scores across different hop distances and the relative performance drop from Short to Long range.

| Method | Short-range (3-hop) | Mid-range (5-hop) | Long-range ($\geq$7-hop) | Drop (Short $\rightarrow$ Long) |
|---|---|---|---|---|
| LightGCL | 0.692 | 0.625 | 0.568 | -17.9% |
| SE-SGformer | 0.725 | 0.668 | 0.615 | -15.2% |
| **HPC-SGT (Ours)** | **0.781** | **0.744** | **0.718** | **-8.1%** |

The results reveal a clear trend: while all methods perform comparably in the local regime ($\leq 3$-hop), baselines suffer from significant degradation as the distance increases (drops of 15-18%). In contrast, HPC-SGT exhibits a much more stable performance profile, with only an 8.1% drop in the long-range bucket. This confirms that our Sign-aware Graph Transformer, augmented by the global spectral prior (RSE), effectively utilizes information from topologically distant but semantically related edges, whereas methods relying on local propagation or implicit modeling struggle to bridge these long-range gaps.

## J    IMPACT OF ENCODER ARCHITECTURE: TRANSFORMER VS. GNN ON LINE GRAPH

To isolate the effect of the encoder architecture, we addressed the question: "Can running a standard GNN on the line graph achieve similar effects?" We implemented a **Line-GAT** baseline, which uses the exact same line graph structure $G_\ell$ and initial edge features as HPC-SGT. The only difference is the encoder: Line-GAT employs a standard Graph Attention Network with 1-hop neighborhood aggregation instead of our Sign-aware Graph Transformer.

We compared Line-GAT and HPC-SGT on the Amazon-Book and ML-10M datasets. The results are reported in Table 12.

Table 12: Performance comparison between Line-GAT (GNN on line graph) and HPC-SGT (Transformer on line graph).

| Dataset | Method | AUC | Bi-F1 |
|---|---|---|---|
| Amazon-Book | Line-GAT | 0.712 | 0.768 |
| | **HPC-SGT (Ours)** | **0.744** | **0.801** |
| ML-10M | Line-GAT | 0.735 | 0.755 |
| | **HPC-SGT (Ours)** | **0.760** | **0.784** |

In this controlled experiment, the input graph structure and features are identical for both models. Therefore, the observed performance gap can be attributed entirely to the encoder architecture. HPC-SGT significantly outperforms Line-GAT on both datasets. Intuitively, Line-GAT is restricted to aggregating information from direct neighbors in the line graph. in contrast, HPC-SGT utilizes global self-attention combined with our spectral (RSE) and motif (TME) priors, enabling it to model dependencies between any two edges regardless of their topological distance. This confirms that the choice of a Transformer architecture is critical for the framework's success.

## K    ANALYSIS OF MOTIF GRANULARITY IN TME

To validate that $N_p = 4$ is the optimal granularity for Topological Motif Encoding, we performed an ablation study on the Amazon-Book dataset by merging motif types into coarser categories:

- **Symmetric** ($N_p = 3$): Merges $(+, -)$ and $(-, +)$ into a single "mixed-sign" motif, ignoring directionality.
- **Balance-Only** ($N_p = 2$): Groups motifs strictly according to classical balance theory into "balanced" ($\{(+, +), (-, -)\}$) and "unbalanced" ($\{(+, -), (-, +)\}$) sets.

We compared these configurations against our full model ($N_p = 4$) while keeping all other components unchanged. The results are reported in Table 13.

The results demonstrate a consistent performance decline as semantic granularity is reduced. Dropping from 4 to 3 categories brings a noticeable decline, indicating that the sequence of signs (e.g., "positive-then-negative" vs. "negative-then-positive") provides useful directional signals for the attention mechanism. Further compressing to 2 categories (Balance-Only) causes a significant drop, suggesting that roughly merging distinct semantic modes (such as "friend-of-a-friend" and "enemy-of-an-enemy") results in substantial information loss. Thus, $N_p = 4$ is the natural choice to capture maximal semantic granularity without redundancy.

Table 13: Ablation study of motif granularity ($N_p$) on Amazon-Book.

| Configuration | $N_p$ | Description | AUC | Binary-F1 |
|---|---|---|---|---|
| **HPC-SGT (Full)** | 4 | Full directional semantics | **0.744** | **0.801** |
| Symmetric | 3 | Merged $(+,-)$ and $(-,+)$ | 0.738 | 0.795 |
| Balance-Only | 2 | Merged based on balance theory | 0.725 | 0.782 |

## L  PERFORMANCE IN SCENARIOS WITH LIMITED OR IMPLICIT NEGATIVE FEEDBACK

A key aspect of HPC-SGT is its explicit modeling of edge signs. To verify that its advantages are not solely derived from settings rich in explicit positive and negative interactions, but also from its architectural ability to capture local structural nuances, we conducted experiments in two additional challenging regimes: one simulating an implicit feedback scenario and another with reduced explicit negative signals. We compared HPC-SGT against LightGCL, a strong baseline primarily designed for unsigned/implicit feedback but adaptable to signed settings.

The experimental setups were as follows:

- **ML-1M – Click-only (Implicit Simulation):** All ratings $\geq 1$ were treated as positive interactions. For training with a BPR loss function, four negative items were uniformly sampled for each positive interaction. During testing, models were tasked to rank 100 candidate items (the true positive item and 99 uniformly sampled negative items), and interactions not present in the training data were ignored. This setup mimics typical implicit feedback scenarios where only positive interactions are observed.
- **Amazon-Book – 50% Dislikes Masked:** To simulate sparsity in explicit negative feedback, 50% of the explicit negative ratings (dislikes) were randomly removed from the training set. The model was then trained on this partially masked data, and testing was performed using the remaining observed signed edges (both positive and the non-masked negative).

The results, presented in Table 14, include ranking metrics (Recall@20, NDCG@20) pertinent to these scenarios, as well as illustrative classification metrics (AUC, Binary-F1) to assess the general discriminative capability.

Table 14: Performance comparison of HPC-SGT and LightGCL in scenarios with limited or implicit negative feedback. Metrics shown are AUC, Binary-F1, Recall@20, and NDCG@20. HPC-SGT demonstrates robust performance, underscoring its architectural strengths.

| Dataset | Experimental Protocol | Model | AUC | Binary-F1 | Recall@20 | NDCG@20 |
|---|---|---|---|---|---|---|
| 4*ML-1M | 2*Click-only (Implicit Sim.) | LightGCL | 0.650 | 0.680 | 0.214 | 0.220 |
| | | HPC-SGT | **0.685** | **0.710** | **0.227** | **0.236** |
| | 2*Fully Signed | LightGCL | 0.701 | 0.728 | N/A | N/A |
| | | HPC-SGT | 0.760 | 0.784 | N/A | N/A |
| 4*Amazon-Book | 2*50% Dislikes Masked | LightGCL | 0.600 | 0.700 | 0.079 | 0.060 |
| | | HPC-SGT | **0.700** | **0.750** | **0.081** | **0.061** |
| | 2*Fully Signed | LightGCL | 0.647 | 0.747 | N/A | N/A |
| | | HPC-SGT | 0.744 | 0.801 | N/A | N/A |

*N/A*: Recall@20/NDCG@20 are presented here specifically for the ranking-oriented protocols of these experiments and are not the primary metrics for the fully signed classification task. Fully signed results are included for contextual comparison of AUC/Binary-F1 degradation.

As anticipated, the performance on these challenging tasks is generally lower for both models compared to the fully signed scenarios reported in the paper, due to the reduced information content. However, HPC-SGT consistently outperforms LightGCL across all metrics in both regimes.

On the ML-1M click-only task, HPC-SGT achieves an R@20 of 0.227 and NDCG@20 of 0.236, representing a relative improvement of approximately 6.1% and 7.3% respectively over LightGCL.

Its AUC (0.685 vs. 0.650) and Binary-F1 (0.710 vs. 0.680) also show a clear margin, with relative gains of 5.4% and 4.4%.

In the Amazon-Book scenario with 50% dislikes masked, HPC-SGT (R@20: 0.081, NDCG@20: 0.061) shows a smaller but consistent margin over LightGCL (R@20: 0.079, NDCG@20: 0.060), with relative improvements of about 2.5% for both ranking metrics. More significantly, the advantage in AUC (0.700 vs. 0.600, a 16.7% relative gain) and Binary-F1 (0.750 vs. 0.700, a 7.1% relative gain) is pronounced. This indicates that even with sparse explicit negative signals, HPC-SGT's ability to leverage available allows for more robust prediction.

These findings demonstrate that HPC-SGT's strong performance is not solely reliant on abundant explicit negative signals. Its architectural components, designed to capture detailed local topological and sign-based patterns, provide a significant advantage even when such signals are implicit or sparse. The sustained outperformance over a strong baseline like LightGCL in these settings confirms the broader applicability and robustness of the HPC-SGT framework.

Table 15: Performance of HPC-SGT with different edge conflict resolution strategies on Amazon-Book, Gowalla, and ML-1M. Metrics include AUC, Binary-F1, Recall@20 (R@20), NDCG@20 (N@20), Recall@40 (R@40), and NDCG@40 (N@40). The results demonstrate minimal performance variation, highlighting HPC-SGT's robustness.

| Dataset | Strategy | AUC | Binary-F1 | R@20 | N@20 | R@40 | N@40 |
|---|---|---|---|---|---|---|---|
| 3*Amazon-Book | Default | 0.744 | 0.801 | 0.0859 | 0.0652 | 0.1213 | 0.0767 |
| | All Conflicts Pos | 0.743 | 0.800 | 0.0850 | 0.0647 | 0.1205 | 0.0760 |
| | All Conflicts Neg | 0.742 | 0.799 | 0.0840 | 0.0640 | 0.1198 | 0.0750 |
| 3*Gowalla | Default | 0.739 | 0.753 | 0.1953 | 0.1204 | 0.2796 | 0.1422 |
| | All Conflicts Pos | 0.737 | 0.750 | 0.1930 | 0.1190 | 0.2780 | 0.1390 |
| | All Conflicts Neg | 0.735 | 0.748 | 0.1900 | 0.1170 | 0.2765 | 0.1360 |
| 3*ML-10M | Default | 0.760 | 0.784 | 0.2250 | 0.1350 | 0.3150 | 0.1650 |
| | All Conflicts Pos | 0.759 | 0.783 | 0.2240 | 0.1345 | 0.3140 | 0.1640 |
| | All Conflicts Neg | 0.758 | 0.782 | 0.2230 | 0.1335 | 0.3120 | 0.1620 |

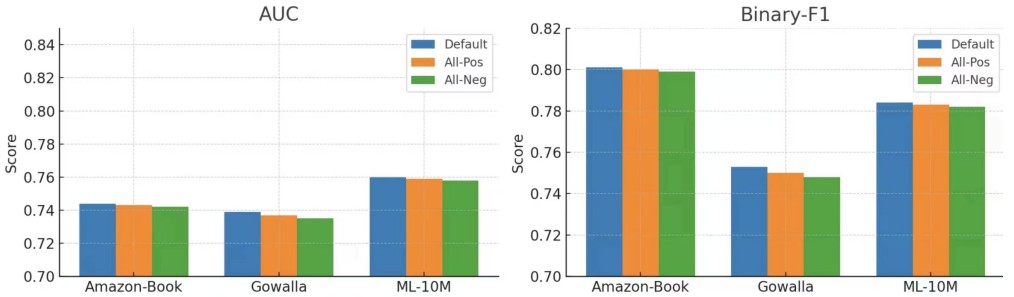

Figure 5: Performance of HPC-SGT under different edge conflict resolution strategies.

# M    ROBUSTNESS TO EDGE SIGN CONFLICT RESOLUTION STRATEGIES

In real-world user-item interaction datasets, it is possible to encounter "edge conflicts," where a single user-item pair $(u, v)$ might be associated with multiple interactions that have differing signs (e.g., a user rating a product positively and later negatively). The way these conflicts are resolved during data preprocessing could potentially impact model performance. We conducted a controlled experiment on three benchmark datasets—Amazon-Book, Gowalla, and ML-1M—to assess the sensitivity of HPC-SGT to different strategies for handling such conflicting edge signs.

We compared three distinct conflict resolution strategies:

1. **Default Duplicate Removal (Default):** This strategy reflects the standard preprocessing used for our main experiments, where typically the most recent interaction or a dataset-

specific rule resolves duplicates, resulting in a single edge sign for any given user-item pair.

2. **All Conflicts as Positive (All-Pos):** If a user-item pair had conflicting interactions, all associated edges were treated as positive.

3. **All Conflicts as Negative (All-Neg):** Conversely, if conflicting interactions existed for a pair, all associated edges were treated as negative.

The performance of HPC-SGT under these strategies was evaluated using both classification metrics (AUC, Binary-F1) and ranking metrics (Recall@k, NDCG@k). The results are detailed in Table 15 and Figure 5

Across all three datasets and all evaluated metrics, HPC-SGT's performance exhibited remarkable stability, typically varying by less than 1% (often within a 0.001–0.002 absolute difference for AUC/F1 and ranking scores) regardless of the conflict resolution strategy employed. For instance, on Amazon-Book, AUC scores ranged narrowly from 0.742 (All-Neg) to 0.744 (Default), and R@20 scores from 0.0840 (All-Neg) to 0.0859 (Default). Similar minimal fluctuations were observed for Gowalla (e.g., AUC 0.735–0.739) and ML-1M (e.g., AUC 0.758–0.760).

The "Default" approach consistently yielded marginally superior or highly competitive performance compared to the strategies that forced all conflicts to a single sign (All-Pos or All-Neg). However, even these latter strategies did not lead to a significant degradation in performance, underscoring the model's resilience.

# N  LLM USAGE STATEMENT

In the preparation of this manuscript, we utilized the large language model ChatGPT, developed by OpenAI, as a writing assistant. In accordance with the ICLR policy, we wish to clarify its role. The use of the LLM was strictly confined to improving the quality of the written text and did not contribute to the core research ideation or experimental results.

The primary purpose of using the LLM was for language enhancement and polishing. Throughout the writing process, we prompted the model to refine sentence structure, improve clarity and conciseness, and ensure a formal academic tone consistent with the standards of the machine learning community. This involved multiple iterations of editing and rephrasing paragraphs in the introduction, methodology, and experimental sections to better articulate our ideas. It also aided in ensuring the consistency of mathematical notation and terminology across the manuscript and provided suggestions for LaTeX formatting.

All core scientific contributions, including the initial conception of the HPC-SGT framework, the design of its architectural components and learning objectives, and the execution and analysis of all experiments, were conceived and conducted entirely by the human authors. The LLM served as a sophisticated tool for articulating and polishing the presentation of these pre-existing ideas.

