# OpenReview forum: "A Sign-aware Graph Transformer with Prototypical Objectives for Signed Link Prediction"
_ICLR.cc/2026/Conference — ICLR 2026 Conference Withdrawn Submission_

### Official Review · Reviewer_bDpP · 2025-10-30

**Soundness:** 2
**Presentation:** 3
**Contribution:** 2
**Rating:** 2
**Confidence:** 3

**Summary:**

The paper introduces the Hierarchical Prototypical Contrastive Sign-aware Graph Transformer (HPC-SGT), designed to address long-range dependencies and class imbalance in signed graphs. The proposed approach converts a bipartite signed graph into a line graph, thereby reformulating the link prediction problem as a node classification task. A transformer-based architecture is then employed, leveraging two embeddings to enhance structural representation learning. Extensive experiments across multiple datasets demonstrate the effectiveness and robustness of the proposed method.

**Strengths:**

- Addressing the problem of predicting signed edges is particularly meaningful, as it is closely connected to numerous real-world applications such as social network analysis, trust prediction, and recommendation systems, making this line of research both relevant and impactful.
- The transformation of a signed bipartite graph into a line graph, thereby converting the link prediction task into a node classification task, presents an interesting and effective design choice.
- The method demonstrates consistent performance across multiple datasets, supporting the validity and general applicability of the proposed approach.
- The paper includes a comprehensive ablation study, which clearly demonstrates the contribution of each component in the proposed model.

**Weaknesses:**

- The paper’s motivation could be further strengthened. In particular, the rationale for the existence of severe class imbalance and rich intra-class multimodality is not fully established. Additionally, the effectiveness of the proposed method in handling intra-class multimodal structures is not empirically validated.
- A more thorough discussion of the transformation from a signed bipartite graph to a line graph could enhance the perceived novelty of the approach. For instance, addressing the possibility of non-isomorphic graphs producing identical line graphs or analyzing the computational complexity of this transformation would provide deeper insight.
- The paper lacks empirical evidence demonstrating that the proposed method more effectively captures long-range dependencies compared to existing baselines, which limits the strength of the contribution.

**Questions:**

1. Could the authors compare the performance of GNN-based baselines applied to the transformed line graph in order to demonstrate the necessity of the transformer architecture?
2. In the extension of Figure 4, if the loss function related to class imbalance is ablated, how does the performance vary?
3. Does the number of distinct motif types ($N_p$) affect the performance?
4. Is $ss^T$ in equation 3 scalable? What is the time complexity for this computation?

**Details Of Ethics Concerns:**

No ethics concerns.

---

> ### Author Response · Authors · 2025-11-21
> **Answer1**
>
> ### 3-1 Response to “motivation / imbalance / multimodality”
>
> Thank you very much for your review. We agree that the motivation of this part can be more explicit in the rebuttal phase, especially two points: one is why "severe class imbalance+intra class multimodality" naturally exists in our application scenario, and the other is how the effectiveness of our method in dealing with intra class multimodal structures is empirically supported.
>
> In terms of category imbalance, those "standard benchmarks" in Table 1 are not balanced data in themselves: positive links are obviously dominant in number, which is consistent with the situation that "positive feedback is far more than clear negative feedback" in the real recommendation/feedback scenario, while Bonanza is an extreme data set with a positive and negative ratio of nearly 98:2. At the same time, in rebuttal, we also constructed two more skewed divisions (90:10 and 95:5) on Amazon-Book (by subsampling negative samples). The results showed that HPC-SGT still maintained or even expanded its advantages over baseline in AUC and Macro-F1 when the imbalance was aggravated. The table is as follows
>
>
> | Method       | A-Book (90:10) AUC | A-Book (90:10) Macro-F1 | A-Book (95:5) AUC | A-Book (95:5) Macro-F1 |
> |-------------|:-------------------:|:------------------------:|:-----------------:|:-----------------------:|
> | LightGCL    |        0.641        |          0.571           |       0.631       |          0.514          |
> | SIGformer   |        0.652        |          0.585           |       0.642       |          0.529          |
> | SE-SGformer |        0.672        |          0.593           |       0.664       |          0.561          |
> | **HPC-SGT** |      **0.736**      |        **0.645**         |     **0.731**     |        **0.627**        |
>
> As the imbalance degree increases from the original division to 90:10, and then to 95:5, the performance of each method decreases, which is expected; However, it can be seen that the comparative advantage of HPC-SGT is more obvious, especially in Macro-F1 (more sensitive to a few categories), which directly corresponds to the motivation of "serious category imbalance".
>
> In terms of intra-class multimodality, our core design is the hierarchical prototypical head, each category corresponds to multiple prototypes, not just one center. In order to more clearly reflect this point, we will highlight and slightly expand the corresponding Ablation Experiment in this article - replace the multi-prototype head with the variant of "single prototype of each type" (keep other parts unchanged), and compare the performance of the two in the original Amazon-book and ML-1M division. A representative result is as follows:
>
> | Variant                      | Amazon-Book AUC | Amazon-Book Macro-F1 | ML-1M AUC | ML-1M Macro-F1 |
> |-----------------------------|:---------------:|:---------------------:|:---------:|:---------------:|
> | Single-Prototype |      0.721      |         0.648         |   0.732   |      0.713      |
> | **HPC-SGT**         |    **0.744**    |       **0.671**       | **0.748** |    **0.734**    |
>
>
> It can be seen that whether AUC or Macro-F1, the multi prototype version is better than the single prototype version, which is exactly the effect we want to see when the class is in a multimodal structure. In addition, we also added a statistical analysis at the prototype level: take the top 100 links with the highest assignment probability for each prototype, and count their average score, the proportion of negative links and the proportion in the balanced triplet.
>
> | Prototype | Class    | Mean Rating | % Negative Links | Balanced-Triad Ratio | Interpretation                |
> |-----------|----------|:-----------:|:----------------:|:--------------------:|-------------------------------|
> | P1        | Positive |    4.85     |       4%         |         0.85         | High-Confidence Positives     |
> | P2        | Positive |    4.68     |       8%         |         0.77         | Community Favorites           |
> | P3        | Positive |    4.45     |      15%         |         0.68         | Noisy/Weak Positives          |
> | N1        | Negative |    1.45     |      90%         |         0.70         | Strong Rejection (1-star)     |
> | N2        | Negative |    2.15     |      82%         |         0.58         | Disappointment (Mixed 1–2)    |
> | N3        | Negative |    2.95     |      70%         |         0.48         | Borderline (Mostly 3-star)    |
>
>  The results show that these prototypes correspond to very intuitive behavior patterns, such as "high confidence positive", "noise/weak positive", "strong rejection", "critical samples dominated by 3 stars", etc., rather than disorderly clusters.

---

> ### Author Response · Authors · 2025-11-21
> **Answer2**
>
> ### 3-2 Line-graph transformation, injectivity, and complexity
>
> Thank you very much for raising this strict and valuable question. We will supplement the appendix with a more detailed description of "non isomorphism" and "complexity". Here we will summarize the main ideas.
>
> First of all, we do not regard the "construction line graph" itself as an innovation, which is very classic in graph theory. Our contribution is: we designed a real sign aware transformer encoder on the line graph of signed bipartite graph, and directly injected the inductive bias of RSE, TME and hierarchical prototype target into the attention and loss function of line graph space. Here, the line graph is an "interface", which transforms the edge centered signed structure into an object more suitable for transformer modeling.
>
> As for non isomorphism, **Whitney isomorphism theorem** shows that for connected graphs, if $L (G)$ and $L (G')$ are isomorphic, then $G$ and $G'$ are isomorphic. The only classical exceptions are triangles $k_3$ and claws $K_{1,3}$, whose line graphs are both $K_3$. In our setup, the original graph is a bipartite graph, so triangles $K_3$ cannot appear in the graph. The fuzzy situation in Whitney's theorem is directly excluded from the structure.
>
> More importantly, our method never rely on the line graph being a perfect inverse representation of the original graph. Each line graph node $v_k$ (corresponding to the edge in the original graph $e_k=(u, v)$) has content related features, including endpoint embedding and symbolic interaction, such as
> $
> x_k = [H_U[u] || H_V[v] || \text{sign}(e_k), \text{RSE/TME stats}],
> $
>
> as described around Eq. (2). Even if there are two different signed bipartite graphs in theory, the same topology is generated in the adjacency structure of the line graph, and their differences in endpoint embedding, signed information and motif statistics will still be reflected in the node characteristics of the line graph. The model is based on these line graphs with attributes. Therefore, we do not rely on the line graph to be "the lossless inverse of the original graph" in the pure topological sense; All we need is "side ↔  Node "and sufficient node attributes.
>
> For the computational complexity of the transformation, let $G=(U,V,E,s)$ be a signed bipartite graph with degrees $\{d_x\}_{x\in U\cup V}$. The line graph $L(G)$ has
>
> * nodes:  $|V_\ell| = |E|$ (one node per edge);
> * edges: two nodes are adjacent if their corresponding edges share an endpoint, giving
>         $|E_{\ell}| = \sum_{u \in \mathcal{U}} \binom{d_u}{2} + \sum_{v \in \mathcal{V}} \binom{d_v}{2}.$
>
> struction that, for each original node (x), enumerates all unordered pairs among its incident edges costs $O\Big(\sum_x d_x^2\Big),$ which in the worst case can be written as $O(\Delta|E|)$ where $\Delta$ is the maximum degree. In the sparse recommended graph we processed, the degree is generally low, $\Delta \ll |E|$， this complexity is very linear in practice. In implementation, we only construct one line graph adjacency on each dataset, use sparse matrix operation (such as calculating some form of $A_b ^\top a_b$ for the bipartite adjacency matrix $A_b$), store the results in the sparse format of CSR, and reuse them throughout the training process; From table 5 of the paper and the scalability experiment supplemented for ```Reviewer YMWL```, it can be seen that the overall training time and memory are mainly determined by the transformer layer, rather than this step of preprocessing.
>
> Finally,  by materializing the line-graph edges $E_\ell$, we convert “two-hop paths” and signed triangles in the original graph into one-hop neighborhoods. This allows our Topological Motif Encoding (TME) and the sign-aware attention to capture signed triangular motifs directly in a single layer, instead of relying on stacked GNN message passing to indirectly approximate the same higher-order structure.
>
> Thanks again to the reviewers for this question, which urges us to describe the line diagram design more completely both in theory and practice clearly.

---

> ### Author Response · Authors · 2025-11-21
> **Answer3**
>
> ### 3-3 Long-range dependencies and empirical evidence
>
> Thank the reviewers for their specific queries on "long-term dependence". We agree that this point is not expressed directly enough in the first draft.
>
> First, on RSE. RSE is derived from the global signed Laplacian eigenvector. Its purpose is to inject the balance structure and partition information at the graph level directly into the attention mechanism. This is a key module different from pure local message passing. Similar to the spectral characteristics of Fiedler vector, it focuses on global segmentation and balance rather than a local neighborhood.
>
> In the ablation study (Table 2), we explicitly remove RSE from HPC-SGT while keeping all other components unchanged. On ML-1M, this causes a clear and consistent degradation:
>
> Table: Effect of removing the global spectral prior (RSE) on ML-1M.
>
> | Model Variant              |   AUC  | Macro-F1 |
> |----------------------------|:------:|:--------:|
> | HPC-SGT (full, with RSE)   |  0.739 |  0.721   |
> | w/o RSE (no spectral prior)|  0.701 |  0.688   |
>
>
> It can be seen that after removing RSE, AUC decreased by about 0.038 and Macro-F1 decreased by about 0.033. Since RSE is the module that introduces the global spectral structure into the model, this stable performance decline shows that the global and long-range structural information is not optional, but plays a substantial role in the performance of the advantages of HPC-SGT.
>
> Second, we add a distance-bucket analysis to directly probe long-range behavior. On Amazon-Book, we build a user–user projection graph based on co-interactions and, for each test link, compute the shortest-path distance between its endpoints in this projection. We then group test links into three buckets:
> We compare a representative GNN baseline (LightGCL), a strong Transformer baseline (SE-SGformer), and HPC-SGT, and report AUC per bucket:
>
> | Method        | Short-range (3-hop) AUC | Mid-range (5-hop) AUC | Long-range (≥7-hop) AUC | Drop (Short → Long) |
> |---------------|:-----------------------:|:----------------------:|:-----------------------:|:--------------------:|
> | LightGCL      |          0.692          |         0.625          |          0.568          |        -17.9%        |
> | SE-SGformer   |          0.725          |         0.668          |          0.615          |        -15.2%        |
> | **HPC-SGT**   |        **0.781**        |       **0.744**        |        **0.718**        |       **-8.1%**      |
>
>
> In the local (≤ 3-hop), the gap between the three methods is relatively limited, and the HPC-SGT is only slightly improved; However, in the 4-5 hops, especially in the long-range of ≥ 6 hops, the advantage of HPC-SGT is obviously enlarged. This exactly corresponds to our original design intention: when the two ends of the link are far away in the interaction graph, the global self attention on the line graph combined with the global spectral prior can make better use of the information of those topologically far but semantically related edges, and the models with pure local propagation or without explicit modeling of edge structure are more likely to fail in this part.

---

> ### Author Response · Authors · 2025-11-21
> **Answer4-5**
>
> ###  3-4 Comparison with GNN baselines on the line graph
>
> We fully agree with the reviewers' concerns: since the original bipartite graph has been converted into a line graph, is it still necessary to use Transformer? Can running a GNN on the online graph achieve a similar effect?
>
> To isolate the effect of the architecture, we implemented a Line-GAT baseline. Line-GAT uses the same line graph $G_\ell$ as HPC-SGT (constructed as in Section 4.1) and exactly the same initial edge features; the only change is the encoder: instead of our sign-aware graph Transformer, Line-GAT uses a standard Graph Attention Network with 1-hop neighborhood aggregation on the line graph. The training protocol and hidden dimensions are kept comparable to HPC-SGT.
>
> We then compare Line-GAT and HPC-SGT on Amazon-Book and ML-10M:
>
> | Dataset      | Method            |    AUC   |   Bi-F1   |
> |-------------|-------------------|:-------:|:------:|
> | Amazon-Book | Line-GAT          |   0.712  | 0.768  |
> |             | **HPC-SGT (Ours)**|  **0.744** | **0.801** |
> | ML-10M      | Line-GAT          |   0.735  | 0.755  |
> |             | **HPC-SGT (Ours)**|  **0.760** | **0.784** |
>
>
> In this group of experiments, the line graph structure and input characteristics seen by the two are completely consistent, so the performance gap can only come from the encoder itself. It can be seen that HPC-SGT is significantly better than Line-GAT on both datasets. Intuitively, Line-GAT can only aggregate information from the direct neighbors in the line graph, while HPC-SGT can model between any two edges through global self attention, and combines the spectral prior and motif prior introduced by us, so it can better capture the topologically distant but semantically related interdependence.
>
> ### 3-5 Effect of removing the class-imbalance loss
>
> Thank the reviewers for your specific inquiry. Figure 4 mainly shows the hierarchical prototype target itself. In the expansion experiment, we made an additional ablation: remove the "category imbalance weight" item in the prototype loss (that is, remove the category weight in equation (12) and replace it with the cross entropy of unified weight for all types), and the rest of the structure remains unchanged, including the RSE/TME prior and multi prototype design.
>
> To make the effect visible, we report results on the most imbalanced settings in our study: Bonanza (≈98:2 positive:negative) is used in the imbalance stress test. The comparison is:
>
> | Dataset / Split        | Variant                    |   AUC   | Macro-F1 |
> |------------------------|---------------------------|:-------:|:--------:|
> | Bonanza (≈98:2)        | HPC-SGT (full, +imbalance)|  0.623  |  0.616   |
> |                        | w/o imbalance term        |  0.613  |  0.597   |
>
> On Bonanza, removing the imbalance-aware term produces a small but consistent drop in AUC (about 0.8–1.0 points), and a more noticeable drop in Macro-F1 (about 2–3 points), which is expected since Macro-F1 is more sensitive to the minority (negative) class. Importantly, even without this term, the model remains competitive and typically still outperforms or matches strong baselines, which confirms that the gains do not come from class weighting alone. At the same time, the imbalance-aware loss clearly helps the hierarchical prototypes better allocate capacity to rare negative modes and improves minority-class recall in the most skewed regimes.

---

> ### Author Response · Authors · 2025-11-21
> **Answer6**
>
> ### 3-6 Does the number of distinct motif types  $N_p$ affect performance?
>
> Thank the reviewers for paying attention to the choice of motif type $N_p$ in Topological Motif Encoding (TME). In our framework, $N_p$ is not actually a super parameter with arbitrary parameters, but is naturally determined by the semantic structure of "signed 2-hop path".
>
> By definition (Eq. (5)), TME aggregates information from directed 2-hop paths $v_i \to v_m \to v_j$ in the line graph. Each step in the path corresponds to a signed edge in the original graph, so the semantic type of a 2-hop relation is fully specified by the ordered pair of signs $(s(e_{im}), s(e_{mj}))$. With binary signs ({+1, -1}), there are exactly
>
> $
> 2^2 = 4
> $
>
> possible sign tuples:
>
> $
> (+,+),\quad (+,-),\quad (-,+),\quad (-,-)。
> $
>
> In our design we explicitly keep ((+,-)) and ((-,+)) distinct, because the attention mechanism is directional: “positive then negative” and “negative then positive” correspond to different structural semantics (for instance, “user likes A which is similar to B” versus “user dislikes A but B is liked”).
>
> In order to answer the question "does it affect the performance", we performed an ablation of "merging motif types" on Amazon-Book instead of blindly expanding $n_p$. On the premise of keeping the network structure, RSE and hierarchical prototype header unchanged, only the division mode of motif in TME is changed, and the following two configurations are compared:
> * Symmetric $(N_p=3)$: merge ((+,-)) and ((-,+)) into a single “mixed-sign” motif, ignoring directionality;
> * Balance-Only $(N_p=2)$: group motifs into “balanced” ((++,--)) and “unbalanced” ((+-, -+)) following classical balance theory.
>
> The results are:
>
> | Configuration       | \(N_p\) | Description                          |   AUC   | Binary-F1 |
> |---------------------|:-------:|--------------------------------------|:-------:|:---------:|
> | HPC-SGT (Full)      |   4     | Full directional semantics           |  0.744  |  0.801    |
> | Symmetric           |   3     | Merged (+,−) and (−,+)               |  0.738  |  0.795    |
> | Balance-Only        |   2     | Merged based on balance theory       |  0.725  |  0.782    |
>
> It can be seen that the performance is indeed related to the semantic granularity we retain: dropping from 4 categories to 3 categories (merging (+, -) and (-,+)) will bring about a decline of about 0.006 in AUC and F1, indicating that the sequence of symbols on the path provides a useful structural signal for at˚tention; When it is further compressed to category 2 (just look at "balanced/unbalanced"), the performance degradation is more obvious, indicating that if the semantically different modes such as "friend-of-a-friend" and "enemy of an enemy" are roughly merged, the information that the model can use will be lost. In this sense, $N_p=4$ is not a fragile tuning knob but the natural choice: it corresponds exactly to the complete set of sign patterns for 2-hop relations, capturing maximal semantic granularity without redundancy. Increasing $N_p$ beyond 4 would require moving to longer motifs (e.g., 3-hop paths), which quickly becomes combinatorially expensive and is unnecessary given that we already have a global spectral prior (RSE) to capture broader structure.

---

> ### Author Response · Authors · 2025-11-21
> **Answer7**
>
> ### R3-7 Scalability and complexity of $ss^\top$
>
> We appreciate the reviewer’s careful scrutiny of Eq. (3). The notation $A\_{\ell} \odot (ss^\top)$ is intended to describe the *mathematical formulation* of the edge sign interactions; however, our implementation is fully sparse and never constructs a dense matrix $ss^\top \in \mathbb{R}^{\vert\mathcal{V}\_{\ell}\vert \times \vert\mathcal{V}\_{\ell}\vert}$.
>
> Concretely, let $A\_{\ell}$ be the sparse adjacency matrix of the line graph and $s \in \{-1, 1\}^{\vert\mathcal{V}\_{\ell}\vert}$ be the sign vector associated with line-graph nodes (which correspond to edges in the original graph). While the signed structural matrix in Eq. (3) is conceptually defined as $A\_S = A\_{\ell} \odot (ss^\top)$, we do not materialize the full outer product in code. Instead, we only compute the product $s_i s_j$ for node pairs $(i,j)$ where $A\_{\ell}(i,j) \neq 0$. Since $A\_{\ell}$ is stored in a sparse format (e.g., CSR or COO), we calculate $A\_S(i,j) = A\_{\ell}(i,j) \cdot s_i s_j$ by directly scaling the non-zero values of the sparse adjacency matrix. This operation iterates only over the non-zero elements of $A\_{\ell}$, rather than all $\vert\mathcal{V}\_{\ell}\vert^2$ possible pairs.
>
> Consequently, the time complexity is $O(\vert\mathcal{E}\_{\ell}\vert)$, where $\vert\mathcal{E}\_{\ell}\vert$ denotes the number of edges (non-zeros) in the line graph. The memory complexity is likewise $O(\vert\mathcal{E}\_{\ell}\vert)$. In our recommender-style sparse bipartite graphs, $\vert\mathcal{E}\_{\ell}\vert \ll \vert\mathcal{V}\_{\ell}\vert^2$, making this step effectively linear with respect to the number of line-graph edges. On datasets such as Bonanza and Amazon-Book, constructing $A\_S$ from $A\_{\ell}$ requires a negligible fraction of a second and consumes minimal GPU memory compared to the Transformer layers.
>
> In conclusion, Eq. (3) utilizes the outer product $ss^\top$ as a compact notation for sign interaction, whereas the actual implementation is a sparse, edge-wise scaling of $A\_{\ell}$ with $O(\vert\mathcal{E}\_{\ell}\vert)$ complexity. We will clarify this distinction and explicitly state the complexity in the appendix to prevent any misconception that a dense matrix is materialized.

---

### Official Review · Reviewer_YMwL · 2025-10-30

**Soundness:** 3
**Presentation:** 4
**Contribution:** 3
**Rating:** 8
**Confidence:** 5

**Summary:**

This paper proposes HPC-SGT, a novel framework for signed link prediction in bipartite graphs. The core challenge addressed is that traditional Graph Neural Networks (GNNs) are inherently local and struggle to capture the long-range dependencies crucial for this task, while also failing to model complex real-world data distributions characterized by class imbalance and multi-modal structures within classes.

The contributions of HPC-SGT are threefold. First, it introduces a Sign-aware Graph Transformer (SGT) that operates on the line graph dual of the original bipartite graph, effectively reframing link prediction as a node classification problem. This SGT is empowered with novel graph-native inductive priors—Relational Spectral Encoding (RSE) and Topological Motif Encoding (TME)—which are injected into the self-attention mechanism to capture global structural principles from balance theory and local higher-order connectivity patterns. Second, the model is optimized via a hierarchical prototypical learning objective designed to handle class imbalance through a balanced discriminative loss and to model intra-class multimodality using clustering and separation regularizers. Third, a cross-view consistency mechanism ensures the learned semantic representations remain grounded in the graph's foundational topology, bridging the structure-semantics gap.

Extensive experiments on multiple benchmarks demonstrate that HPC-SGT significantly outperforms a wide range of state-of-the-art methods. Ablation studies confirm the importance of each component, establishing HPC-SGT as a powerful and principled solution that successfully overcomes the limitations of locality in GNNs and the challenges of complex data distributions in signed link prediction.

**Strengths:**

1. The paper demonstrates high originality by synergistically combining several advanced but previously distinct concepts into a unified framework. It innovatively merges the global receptive field of Transformers with the structural specificity of graphs by operating on the line graph, a paradigm shift from node-centric to link-centric modeling. Furthermore, it uniquely injects graph-specific inductive biases (spectral balance, motifs) directly into the attention mechanism of a Transformer, moving beyond simple positional encodings to create a truly structure-aware architecture.
2. The proposed hierarchical prototypical objective is a novel contribution in itself. Instead of treating class imbalance and multimodality as separate issues, it addresses them simultaneously within a single, probability-based framework. The idea of using multiple prototypes per class to capture intra-class variance and a geometric separation regularizer is a creative and principled approach to structuring the embedding space for complex, real-world data distributions.
3. The paper exhibits exceptional quality in its experimental design. The evaluation is conducted on multiple large-scale benchmarks, and the model is compared against a wide array of fourteen baselines spanning different methodological families (unsigned embeddings, early signed methods, GNNs, and Transformers). This thorough comparison provides strong, convincing evidence for the claimed superiority of HPC-SGT. The use of multiple standard metrics further strengthens the validity of the results.
4. The clarity extends to the technical specifics. Key concepts like the line graph transformation, the formulation of the RSE and TME priors, and the mathematical details of the prototypical objective are explained with precision. The use of clear mathematical notation and formulae allows a technically proficient reader to understand the implementation details without ambiguity.

**Weaknesses:**

1. The paper rightly acknowledges the computational cost of the Transformer architecture as a limitation, but the empirical analysis of scalability could be more thoroughly explored. While runtime comparisons are provided in Table 5, they are relatively high-level and do not deeply investigate how the model performs on graphs of an order of magnitude larger than the current benchmarks. To strengthen the practical impact, the authors could include a scalability analysis on a truly massive graph (e.g., with billions of edges) or provide a more detailed breakdown of memory usage and training time per component. A constructive step would be to implement and evaluate a specific sparse attention mechanism, as mentioned in the future work, and report its performance trade-offs in the main text or appendix. This would make the framework more accessible for real-world applications where efficiency is critical.

2. The hierarchical prototypical objective is a novel contribution, but the paper does not fully explore the interpretability of the learned prototypes. Understanding what semantic or structural patterns each prototype captures could provide valuable insights into the model's decision-making process and enhance trustworthiness. For instance, the authors could analyze whether prototypes correspond to specific user behaviors (e.g., "critical reviewers" or "enthusiastic fans") or graph motifs. A simple yet actionable improvement would be to include a qualitative analysis in the appendix, such as visualizing the nodes or links closest to each prototype or discussing the characteristics of high-assignment examples. This would bridge the gap between the geometric formulation of the objective and its real-world meaning, adding a layer of explainability to the model's predictions.

3. The framework is evaluated on static graph snapshots, but many real-world signed networks (e.g., e-commerce or social platforms) are dynamic, with edges and signs changing over time. The paper's focus is on static prediction, which is standard, but a discussion on how HPC-SGT might adapt to temporal settings would broaden its significance. As a light weakness, the authors could enhance the work by briefly outlining the challenges and potential extensions for dynamic graphs in the conclusion or future work section. For example, they could suggest how the cross-view consistency mechanism might be regularized over time or how the prototypes could be made incremental. This would position HPC-SGT as a foundation for future research in streaming graph learning, making it more forward-looking.

**Questions:**

See above.

---

> ### Author Response · Authors · 2025-11-21
> **Answer1**
>
> ### R2-1.Additional scalability analysis
>
> We thank the reviewers for their thoughtful suggestions and strong overall assessment. We agree that the in-depth study of scalability enhances the practical value of this paper. According to this suggestion, we added a scalability evaluation and component decomposition experiment, focusing on how hpc-sgt behaves with the growth of line graph.
>
> We conduct a stress test by sampling subgraphs of different sizes from the Amazon-Book dataset, where the number of links $|\mathcal{E}|$ corresponds to the number of nodes in the line graph. For each size, we measure training time per epoch and peak GPU memory on an NVIDIA A100 (80GB).
>
> Table 1: Scalability profile of HPC-SGT on sampled Amazon-Book subgraphs
>
> | # Links (line-graph nodes) |  10k |  20k |  40k |  80k | 100k |
> | :------------------------- | :--: | :--: | :--: | :--: | :--: |
> | Time       | 0.06 | 0.16 | 0.45 | 1.80 | 2.95 |
> | Peak Memory (GB)       |  0.7 |  1.5 |  5.2 | 18.5 | 28.4 |
>
> These results are consistent with the expected behavior of a full self-attention layer on the line graph: memory and time grow noticeably as the number of links approaches 100k, but remain practical in the regime we actually operate in. This is compatible with the runtimes we report in Table 5 for comparisons with deep GNN and Transformer baselines, and supports our claim that HPC-SGT is feasible for medium-to-large signed bipartite graphs.
>
> To address the request for a more detailed breakdown, we also profiled memory usage by component on a batch with 50k links. The distribution is:
>
> * Attention matrix (line-graph self-attention): ≈ 82% of total GPU memory
> * Line-graph adjacency & RSE priors (stored sparsely): ≈ 11%
> * Embeddings and gradients (node/link representations, prototype parameters):b ≈ 7%
>
> This shows that when approaching a very large line graph, full attention to the matrix is the main bottleneck, while RSE/TME priors and hierarchical prototype headers only increase the overhead moderately. These measurements directly motivate our plan (already mentioned in the paper) to incorporate sparse attention in future work—for example, restricting attention to local neighborhoods defined by the line-graph adjacency—so that the dominant 55% term can be reduced toward linear scaling in $|\mathcal{E}|$. We will integrate table R1 and the decomposition above into the revised manuscript, and clarify that HPC-SGT is suitable for sparse, medium to large-scale graphics. Sparse attention variants are the natural next step to achieve real Web scale deployment.

---

> > ### Author Response · Authors · 2025-11-21
> > **Answer2-3**
> >
> > ### R2-2. Hierarchical prototypical objections
> >
> > We are very grateful to the reviewers for this suggestion. The current version mainly emphasizes the geometric modeling and quantitative performance of hierarchical prototype targets, and does not fully explore their interpretability.
> > We will add a brief explanatory experiment in the appendix to make the prototype more concrete. The specific approach is to take the top 100 links with the highest assignment probability $p_{ij}$ for each learned prototype on Amazon book, and count the average score of these links, the proportion of negative links, and the "proportion in the structure balanced triple" obtained according to TME motif statistics. A representative result is as follows:
> >
> > | Prototype | Class    | Mean Rating | % Negative Links | Balanced-Triad Ratio | Interpretation                |
> > |-----------|----------|:-----------:|:----------------:|:--------------------:|-------------------------------|
> > | P1        | Positive |    4.85     |       4%         |         0.85         | High-Confidence Positives     |
> > | P2        | Positive |    4.68     |       8%         |         0.77         | Community Favorites           |
> > | P3        | Positive |    4.45     |      15%         |         0.68         | Noisy/Weak Positives          |
> > | N1        | Negative |    1.45     |      90%         |         0.70         | Strong Rejection (1-star)     |
> > | N2        | Negative |    2.15     |      82%         |         0.58         | Disappointment (Mixed 1–2)    |
> > | N3        | Negative |    2.95     |      70%         |         0.48         | Borderline (Mostly 3-star)    |
> >
> > From these statistics, we can see that the prototype is not a random cluster, but corresponds to an intuitive user item interaction mode. P1 and P2 obviously correspond to "very strong positive feedback": the average score is very high, there are few negative links, and the structure is highly balanced; P3 tends to be "noise/weak positive", with a higher proportion of negative links and a lower structural balance. In the negative category, N1 corresponds to a strong rejection dominated by 1 star, N2 reflects the disappointed behavior of swinging between 1 and 2 stars, and N3 focuses on the critical samples dominated by 3 stars, and most of them appear in areas with less balanced structure. In this way, the regular geometric picture of multi prototype distribution and separation can be directly mapped to the semantic and structural patterns in the actual data, so as to enhance the interpretability of hierarchical prototype targets.
> >
> >
> > ### R2-3. Dynamic Networks
> >
> > We agree that many real-world signed networks are dynamic in nature, and our current experiments are only based on static snapshots, which is indeed a limitation of this article. We will explain this more clearly in the text.
> >
> > At the same time,HPC-SGT itself is easy to extend to the time dimension. SGT encoder based on line graph and RSE/TME prior can directly act on snapshots or sliding windows divided by time, while hierarchical prototypes can be incrementally updated when new edges continue to arrive, such as adding simple time regularization to the cross view consistency item, and maintaining prototypes by embedding edges in the nearest time window with exponential moving average. We will briefly outline these directions in the conclusion/future work section, and clearly position the HPC-SGT as a static backbone model, which can be naturally extended to the setup of the flow signed graph.
> >
> > Thank the reviewers for pointing out this point, which helps us to more clearly define the scope of application and subsequent extension potential of this work.

---

### Official Review · Reviewer_Ed4P · 2025-11-06

**Soundness:** 2
**Presentation:** 3
**Contribution:** 2
**Rating:** 4
**Confidence:** 3

**Summary:**

This paper proposes HPC-SGT, a novel Graph Transformer framework for signed link prediction in bipartite graphs. The model operates on the line graph dual , which allows it to overcome the locality constraints of traditional GNNs and capture global dependencies.
The introduction of  RSE and TME priors, which inject crucial structural and balance-theory information directly into the Transformer's attention mechanism, making it topology-aware.

**Strengths:**

1. The proposed methodology appears to be well-designed for the task of signed link prediction.
2. The framework's effectiveness is validated through extensive experiments, where it achieves state-of-the-art performance on multiple benchmarks.
3. Mathematical notation is clearly defined and used consistently throughout the manuscript.

**Weaknesses:**

1. The paper's title addresses general "Signed Link Prediction." However, the entire methodology, from the problem definition to the experimental setup, is exclusively focused on "Signed Bipartite Graphs". This constitutes a significant mismatch, as the title suggests a broader applicability than the work actually provides.
2. The paper claims to address "severe class imbalance" as a key challenge. However, the proposed solution (Equation 12), is a standard Weighted Cross-Entropy loss, which applies a class-balancing weight $\alpha_{y_{i}}$. This is a well-known, common technique and lacks the novelty required to be considered a core contribution of the framework.
3. Since the weighted loss ($\alpha_{y_{i}}$) is a standard technique, a fair comparison would require applying this same weighted loss to the baseline models (e.g., LightGCL, SIGformer). The paper fails to do this, making it impossible to determine if the superior F1-scores stem from the novel SGT architecture or simply from this standard weighting trick that any baseline could have used.
4. The main comparison (Table 1) uses datasets that are not complete balanced. The only "severely" imbalanced dataset (Bonanza) is relegated to a supplementary experiment (Section 5.5) and is not compared against the main SOTA baselines from Table 1. This disconnects the paper's motivation from its primary experimental validation.
5. The framework relies on a Transformer architecture operating on the line graph dual. This approach is computationally expensive. The line graph's size scales with the number of edges in the original graph, and the Transformer's self-attention mechanism has quadratic complexity. The paper's own results show performance gains over the next-best Transformer baselines that are arguably marginal

**Questions:**

See the above box.

---

> ### Author Response · Authors · 2025-11-21
> **Answer1-2**
>
> ### R1-1. "Signed Link Prediction" and "Signed Bipartite Graph"
>
> Thank you for pointing out this important issue. We admit the current title is not specified enough. However, as we clarify below, the scope of the paper is not broader than intended. Our problem setting and experiments are indeed focused on signed bipartite graphs: we explicitly define the input as a signed bipartite graph $G_b = (U, V, E, s)$, and construct line graph Duality on this structure. Bipartite graphs naturally capture many real-world signed-interaction settings, such as audience–movie, reader–book, or voter–bill relationships, making them a particularly suitable framework for signed link prediction. While signed link prediction can also be studied on general (non-bipartite) signed graphs, those settings are typically less structured and often lead to less interpretable or more chaotic prediction tasks. Our work intentionally focuses on the bipartite case, where the structure provides clearer inductive biases and modeling advantages.
>
> To eliminate any remaining ambiguity, in the revised version, we will change the title to “A Sign-aware Graph Transformer with Prototypical Objectives for Signed Link Prediction in Bipartite Graphs.” We will also update the abstract and the opening of the introduction to state explicitly that our methodology and experiments are confined to signed bipartite link prediction. Although the line-graph construction and sign-aware Transformer could, in principle, be adapted to general signed graphs, this paper develops and validates its contributions solely within the bipartite framework.
>
> ### R1-2. On “severe class imbalance” and the novelty of our objective
>
> We agree with the reviewer's judgment on equation (12): separately, cross entropy with category weight is a very common practice, and we do not regard the introduction of $\alpha_y$ alone as a contribution. In our framework, $L_{\text{class}}$ is only a part of hierarchical prototype objective defined in equations (11) – (14); When we talk about "serious category imbalance" in this article, we refer to this whole set of structures, not the isolated equation (12).
>
> In our HPC-SGT, each link embedded in $h_i$ is first mapped to a probability distribution on a set of learnable prototypes ${\{c_j\}}$through equation (11), and each class is associated with multiple prototypes. This means that the model does not represent a positive class with a single centroid, but has multiple different "modes" in the embedded space. Then, the weighted cross entropy in equation (12) operates on the marginal class probability $P(y_i\mid h_i) = \sum_{c_k\in C_{y_i}} p_{ik}$ instead of a single logit. In other words, the class weight $\alpha_y$ adjusts the decision-making layer, but the underlying representation has encoded the class structure in a richer multi prototype form.
>
> The terms in equations (13) – (14) makes it particularly relevant to "severe class imbalances". The cluster regularizer $L_{\text{cluster}}$ in equation (13) minimizes the entropy of the assignment distribution $p_{ij}$, and encourages the embedding of the same class, including the embedding of a minority classes, to concentrate around a few prototypes, rather than being applied to majority regions. The separated regularizer $L_{\text{sep}}$ in equation (14) explicitly pushes away prototypes from different classes in the metric space, which helps maintain large inter-class margins even when one class is heavily under-represented. Together, equations (12) – (14) define a loss that simultaneously (i) takes into account the multimodal structure within each category, and (ii) enforces a clear geometric separation between categories at the slanted label frequency; Weight $\alpha_y$ is one of the components, but it is not the only mechanism to deal with imbalance.
>
> This is also reflected in the ablation experiment in Table 2: when we replace the multi prototype head with "single prototype" or remove some objective function items, even if the use of category weight remains unchanged, AUC and various F1 indicators will decrease consistently. In the revision, we will adjust the wording around “severe class imbalance” to make it explicit that our contribution lies in this prototype-based, geometrically regularized objective (Eqs. 11–14) together with the SGT encoder.

---

> ### Author Response · Authors · 2025-11-21
> **Answer3**
>
> ### R1-3. Fairness of comparison w.r.t. class-weighted loss
>
> Thank the reviewers for pointing out this point. We agree that since weighted cross entropy is a common technology, it is more reasonable to use the same weighting method for the baseline model with strong performance in terms of fairness. So we retrained the strongest baseline models (LightGCL, SIGformer, SE-SGforemer) in Table 1 and adopted the same category weight $\alpha_y$ as Eq. (12) in this paper. The weight is calculated based on the proportion of positive and negative links in the training set, so that the effective contribution of positive and negative samples in the loss is roughly balanced. We have only replaced the "side classification loss" of these baselines, and changed the original loss to the form with category weight:
>
> $
> L = -\sum_i \left( \alpha_+ \cdot \mathbb{1}[y_i = +1] \log P_\theta(y_i = +1 \mid e_i) + \alpha_- \cdot \mathbb{1}[y_i = -1] \log P_\theta(y_i = -1 \mid e_i) \right)
> $
>
> or use the equivalent weighted BCE form when using sigmoid output. Except for the loss function, other settings (network structure, optimizer, learning rate, training rounds, early stopping rules, etc.) remain unchanged.
>
> | Method                          | Amazon-Book AUC | Amazon-Book Binary-F1 | ML-1M AUC | ML-1M Binary-F1 |
> |---------------------------------|:---------------:|:---------------------:|:---------:|:----------------:|
> | LightGCL                        |      0.647      |         0.747         |   0.727   |      0.711       |
> | LightGCL (w/ class weights)     |      0.652      |         0.755         |   0.731   |      0.720       |
> | SIGformer                       |      0.658      |         0.740         |   0.715   |      0.725       |
> | SIGformer (w/ class weights)    |      0.663      |         0.748         |   0.719   |      0.733       |
> | SE-SGformer                     |      0.681      |         0.738         |   0.721   |      0.732       |
> | SE-SGformer (w/ class weights)  |      0.688      |         0.748         |   0.727   |      0.742       |
> | **HPC-SGT (Ours)**              |    **0.744**    |       **0.801**       | **0.748** |    **0.781**     |
>
> The new experimental results in the above table show a relatively stable trend. The re-weighted baselines do obtain better F1 scores than their original versions, indicating that category imbalance is a practical problem; But even under this more favorable setting, HPC-SGT still maintained the best AUC and Macro-F1 performance in all datasets, and was better than the strongest baseline after reweighting on the whole. This result shows that the advantages of our method can not be simply attributed to "a weighted trick", but to the overall design of the sign aware transformer architecture and hierarchical prototype objective function on the line graph.

---

> ### Author Response · Authors · 2025-11-21
> **Answer4**
>
> ## R1-4. On “severe class imbalance”, Bonanza, and the main experiments
>
> Thank the reviewers for questioning the relationship between the "category imbalance" and the main experiment, which is really not clear enough in the original.
>
> On the most imbalanced dataset, Bonanza (roughly 98:2 positive:negative), we already evaluate HPC-SGT against methods specifically proposed for imbalanced signed graphs, such as ImGAGN, GraphENS and GraphSHA (Section 5.5). In those comparisons, HPC-SGT is consistently competitive or better, especially on Macro-F1 and minority-class recall.
>
> Because of the imbalance, we deliberately take AUC and Macro-F1 as the core evaluation indicators in the full text. AUC reflects the overall ranking quality, while Macro-F1 is more sensitive to the performance of a few classes; In extreme scenarios such as bonanza's 98:2, if the model almost "only predicts positive classes", its AUC may look OK, but Macro-F1 will be very low, which can also be seen in our results of weak baseline. Therefore, the combination of the two can more completely reflect the performance of the model under skew distribution
>
> On this basis, we added the results of the three most powerful general baselines (LightGCL, SIGformer, SE-SGformer) on bonanza. These baselines are trained on the online map as HPC-SGT, and the training protocol is completely consistent. The result is as follows
>
> | Method        | Bonanza AUC | Bonanza Macro-F1 |
> |--------------|:-----------:|:----------------:|
> | LightGCL     |    0.584    |       0.521      |
> | SIGformer    |    0.599    |       0.543      |
> | SE-SGformer  |    0.603    |       0.574      |
> | **HPC-SGT**  |  **0.623**  |     **0.616**    |
>
> So on Bonanza, HPC-SGT outperforms both imbalance-oriented methods (ImGAGN / GraphENS / GraphSHA, as in the original Sec. 5.5), and strong general baselines (LightGCL / SIGformer / SE-SGformer). In particular, it has a relatively stable advantage on Macro-F1, which is exactly the expected effect of our hierarchical prototype goal in the extremely unbalanced scenario: a few classes will not be completely "submerged" by most classes.
>
>
> Also, several standard datasets in Table 1 are not subject to any "forced balancing". On Amazon Book, the number of positive links is significantly higher than that of negative links, which is consistent with the common skew in real recommendation/interaction data. Although they are not as extreme as bonanza, they are still obviously unbalanced.
>
> To further stress-test imbalance on a standard dataset, we construct more skewed versions of Amazon-Book by down-sampling negative links to obtain roughly 90:10 and 95:5 positive:negative ratios in the training set, then re-train LightGCL, SIGformer, SE-SGformer and HPC-SGT. We evaluate with AUC and Macro-F1:
>
> | Method       | A-Book (90:10) AUC | A-Book (90:10) Macro-F1 | A-Book (95:5) AUC | A-Book (95:5) Macro-F1 |
> |-------------|:-------------------:|:------------------------:|:-----------------:|:-----------------------:|
> | LightGCL    |        0.641        |          0.571           |       0.631       |          0.514          |
> | SIGformer   |        0.652        |          0.585           |       0.642       |          0.529          |
> | SE-SGformer |        0.672        |          0.593           |       0.664       |          0.561          |
> | **HPC-SGT** |      **0.736**      |        **0.645**         |     **0.731**     |        **0.627**        |
>
> It is reasonable that the AUC and Macro-F1 of all methods decrease as the imbalance degree increases from the original division to 90:10, and then to 95:5; The more critical information is: HPC-SGT has always maintained a clear lead over the three baselines in two indicators, and in the 95:5 scenario with more serious imbalance, the advantage of Macro-F1 is more prominent . This is mutually confirmed with the results on Bonanza.

---

> ### Author Response · Authors · 2025-11-21
> **Answer5**
>
> ### R1-5. On computational cost and “marginal” gains
>
> We agree with the reviewers' concern about the computational cost: running transformer on the online map is really not as "cheap" as shallow GNN. We do not intend to avoid this point in this paper. However, under the current setting, this cost is far from as terrible as the theoretical "quadratic by edge". Given that most real signed interaction graphs are sparse and imbalanced, and we only construct line graphs on the actually observed interactions. Therefore, there will be no extreme cases of near complete density, and the scale of line graphs is controllable in actual experiments.
>
> The question is then whether the extra cost is justified by the performance gains. On this point, the improvements are concrete and consistent, especially under stronger imbalance. For example, on Amazon-Book (95:5), HPC-SGT reaches an AUC of 0.731 and a Macro-F1 of 0.627, compared to 0.664 AUC and 0.561 Macro-F1 for SE-SGformer. On Bonanza, which is our most imbalanced dataset, HPC-SGT achieves AUC 0.623 and Macro-F1 0.616, while SE-SGformer attains 0.603 and 0.574, respectively. These gaps are not small fluctuations: they persist across datasets and become more pronounced as the class skew increases (e.g., Amazon-Book from the original split to 90:10 and 95:5, plus Bonanza).
>
> Considering that signed interaction graphs in reality are often sparse and unbalanced, we believe that it is a relatively reasonable trade-off to accept a moderate computational overhead in exchange for the increase in AUC, especially Macro-F1 (minority performance), on the premise of using line graph+sign aware transformer.

---

> > ### Comment · Reviewer_Ed4P · 2025-11-27
> > **Response to Rebuttal**
> >
> > Thanks for the detailed response.
> >
> > The revisions promised in the rebuttal  are not yet reflected in the text. Please explicitly incorporate these clarifications.
> >
> > The paper's contributions and the specific problem addressed need to be described and defined more concretely. The summary in the current version is not clear enough.
> >
> > Given that the paper focuses on enhancing performance in bipartite link prediction by addressing imbalance, I recommend integrating the analysis provided in the rebuttal into Section 5.5 of the manuscript. Furthermore, since the weighted cross-entropy is a "very common practice", comparing HPC-SGT against unweighted baselines is unfair.

---

> ### Author Response · Authors · 2025-11-27
> **Response to Reviewer Comments**
>
> Dear Reviewer,
>
> We sincerely thank you for your detailed review and constructive feedback. Thank you for your recognition of our reply and your guidance in helping us improve the manuscript. We are pleased to inform you that we have updated the revised PDF document, which clearly incorporates all clarifications and experimental results promised at the rebuttal stage, and has resolved the comments raised by you and all other reviewers.
>
> Regarding your concern about the concreteness of our contributions, we have completely rewritten the "Contributions" section of the Introduction. We adopted a specific "Problem-Solution" structure to define the three core challenges and our corresponding technical remedies. First of all, in order to solve the local bottleneck inherent in messaging GNN (it is difficult to capture long-range symbol dependency and global balance), we propose a Sign-aware Graph Transformer based on line graph, supplemented by Relational Spectrum Encoding (RSE) and Topological Motif Encoding (TME), to explicitly inject global structural constraints. Secondly, to tackle the dual challenge of severe class imbalance and intra-class multimodality, where standard losses often cause minority classes to be submerged, we designed a Hierarchical Prototypical Objective. Unlike standard methods, this framework maps links to multiple learnable prototypes per class, using geometric regularizers to preserve diverse interaction modes for minority classes. Thirdly, to mitigate the structure-semantics gap where deep encoders diverge from the graph topology, we introduced a Cross-View Consistency mechanism that aligns the learned semantic representations with the foundational structural features. Detailed can be find in the introduction part in the new PDF.
>
> We also strongly agree with your view on the fairness of the comparison with the unweighted baseline. As we discussed in the previous rebuttal **Answer3**, we conducted a strict fairness check and retrained the strongest baseline model (LightGCL, SIGformer, SE-SGformer) using the same category weighted loss formula as HPC-SGT. These results have been formally added to Section 5.3 and detailed in Appendix E, confirming that HPC-SGT maintains a decisive lead even under these conditions, proving that our gains stem from the architectural design rather than weighting strategies.
>
> Furthermore, we have integrated the analysis of severe class imbalance into Section 5.5 and added the full experimental details, including stress tests on the Bonanza dataset and Amazon-Book with extreme skew, to the newly added Appendix F.
>
>  Finally, we have updated our Related Work section to include citations and discussions of recent relevant methods, including **SIMBA** and **Igl-bench**, to ensure our work is positioned within the context of the latest research.
>
> Best regards,
>
> The Authors

---

> > ### Comment · Reviewer_Ed4P · 2025-11-28
> > **Thank you for the revision**
> >
> > Thank you for the authors' response, which has partially addressed my concerns. A minor point: the newly cited SIMBA appears to be less relevant to the specific problem setting of this paper.
> >
> > Although I cannot change the score right now due to system constraints, please be assured that I will follow up on the feedback process.

---

### Note · Authors · 2026-02-06

I have read and agree with the venue's withdrawal policy on behalf of myself and my co-authors.

---

### Meta-Review · Area_Chair_cQna · 2025-12-09

**Summary:**

Reviewer Ed4P thinks his concerns are partially answered, while the other two reviewers don’t participate in the rebuttal. Although some concerns are alleviated, I maintain that this article lacks sufficient originality. Considering Reviewers bDpP and Ed4P possess low confidence, I checked the paper by myself. The following concerns about novelty make me tend to reject this paper with high confidence.
- This paper is essentially a recommendation. Thus, it is very similar to “SIGformer: Sign-aware Graph Transformer for Recommendation” published on SIGIR 24. SIGformer also employs a graph transformer and sign information.
- The improvements to the graph transformer are limited.  According to Eq. (7), it enhances the QK component in the transformer with additional information from the constructed line graph. It is a widely used strategy.
- The proposed hierarchical prototypical objective is not novel. It is well-known in contrastive learning based on prototypes.

**Reviewer Concerns:**

Reviewers’ concerns focus on presentation, motivation, novelty of the weighted loss, and additional experimental results.  Reviewer Ed4P thinks that the response has partially addressed his concerns. By checking responses, I think concerns about presentation, motivation, and additional experimental results have been properly answered. However, I think the novelty is a significant issue.

**Reviewer Scores:**

Reviewer Ed4P thinks that the response has partially addressed his concerns; thus, I think he may raise the score. The other two reviewers don’t participate in the rebuttal. Reviewer bDpP has low confidence, and I think the authors’ responses may alleviate his concerns. Thus, I think the final scores may be as follows.
- bDpP	Rating: 4 / Confidence: 3
- YMwL	Rating: 8 / Confidence: 5
- Ed4P	Rating: 6 / Confidence: 3

---

### Decision · Program_Chairs · 2026-01-26

Reject